# TARO: TOWARD SEMANTICALLY RICH OPEN-WORLD OBJECT DETECTION

## ABSTRACT

Modern object detectors are largely confined to a "closed-world" assumption, limiting them to a predefined set of classes and posing risks when encountering novel objects in real-world scenarios. While open-set detection methods aim to address this by identifying such instances as *Unknown*, this is often insufficient. Rather than treating all unknowns as a single class, assigning them more descriptive subcategories can enhance decision-making in safety-critical contexts. For example, identifying an object as an *Unknown Animal* (requiring an urgent stop) versus *Unknown Debris* (requiring a safe lane change) is far more useful than just *Unknown* in autonomous driving. To bridge this gap, we introduce TARO, a novel detection framework that not only identifies unknown objects but also classifies them into coarse parent categories within a semantic hierarchy. TARO employs a unique architecture with a sparsemax-based head for modeling objectness, a hierarchy-guided relabeling component that provides auxiliary supervision, and a classification module that learns hierarchical relationships. Experiments show TARO can categorize up to 29.9% of unknowns into meaningful coarse classes, significantly reduce confusion between unknown and known classes, and achieve competitive performance in both unknown recall and known mAP. Code is available at: `https://anonymous.4open.science/r/TARO`

## 1 INTRODUCTION

Object detection, which is a critical task in computer vision that involves both localizing and classifying objects, has evolved significantly over the years. Early approaches relied on hand-designed features such as Haar-like features (Viola & Jones, 2001) and Histograms of Oriented Gradients (HOG) descriptors (Dalal & Triggs, 2005). They were later gradually replaced by deep learning-based approaches, which can be further divided into two-stage methods (Girshick et al., 2014; Lin et al., 2017a; Ren et al., 2015; Girshick, 2015) and one-stage methods (Redmon et al., 2016; Liu et al., 2016; Law & Deng, 2018; Lin et al., 2017b). More recently, vision-transformer-based detectors, pioneered by DEtection TRansformer (DETR) (Carion et al., 2020), have gained significant popularity. By leveraging the attention mechanisms in transformers, these models are able to capture long-range dependencies and global context.

Despite these advancements, a key challenge persists: most detectors can only recognize the classes defined in their training datasets. This limitation stems from the **closed-world assumption**, which presumes that all objects encountered during deployment are already seen and learned during training. Such an assumption is not only unrealistic but can also pose safety risks in practical applications. For instance, an autonomous vehicle trained on pedestrians and cars can fail to detect out-of-distribution (OOD) objects (e.g., e-scooters or construction barriers), causing unsafe behavior.

To address this safety gap, recent research on open-world object detection (OWOD) (Joseph et al., 2021; Wu et al., 2022b; Yu et al., 2022; Wu et al., 2022a; Gupta et al., 2022; Zohar et al., 2023; Sun et al., 2024; He et al., 2024; Luo & Zhou, 2024; Xi et al., 2024) has focused on detecting OOD objects and label them as *Unknown*. However, simply localizing unknown objects is not sufficient. When encountering something unfamiliar, humans rarely assign a generic *Unknown* label but instead categorize it into broader, semantically meaningful groups. For instance, at a fruit market, an unfamiliar object resembling an orange would more likely be regarded as an *Unknown Fruit* rather than just an *Unknown Object*. Applying this idea to object detection can be highly beneficial. If an

autonomous vehicle detects an e-scooter ahead as an *Unknown Vehicle*, the system can infer that the object might exhibit potential movement and therefore wait. In contrast, if a construction barrier is detected and categorized as an *Unknown Obstacle*, the planner should instead generate a bypass trajectory, since the object is expected to remain stationary.

Building on this idea, we propose incorporating a hierarchical taxonomy of objects to derive more information of unknown objects. The proposed framework, TARO, is designed to localize both known and unknown objects. Known objects are classified into their fine-grained categories, represented by the leaf nodes of the taxonomy, while unknown objects are assigned to coarser categories at higher levels of the hierarchy.

Our main contributions are as follows:

- We extend the standard OWOD setting by introducing the task of categorizing unknown objects into meaningful coarse categories, rather than treating them as a single undifferentiated class.
- We present TARO, which integrates (i) a sparsemax-based objectness head for predicting whether a bounding box encloses a valid object, independent of category, (ii) a hierarchy-aware classification module enforcing consistent predictions across taxonomic levels, and (iii) hierarchy-guided relabeling for auxiliary supervision of objectness.
- We conduct extensive experiments on multiple OWOD benchmarks, showing that TARO achieves high unknown recall, effectively categorizes unknown objects into coarse categories, and significantly reduces confusion between known and unknown objects.

## 2 PRELIMINARY

**DETR-based Detectors.** As the first fully end-to-end object detection framework, DETR (Carion et al., 2020) combined a convolutional neural network (CNN) backbone with a transformer architecture. By reformulating detection as a set prediction problem and leveraging bipartite matching loss, DETR eliminated traditional hand-crafted components such as anchor generation and non-maximum suppression (NMS).

The DETR architecture in general consists of four main components: (1) a CNN backbone that extracts feature maps from the input image; (2) a transformer encoder–decoder module, where the encoder processes the feature map into a global, contextualized representation, and the decoder employs a fixed set of learned object queries to attend to this representation, producing object-level embeddings for each query; (3) prediction heads that transform each query embedding into a class probability and a bounding box prediction; and (4) a Hungarian matcher that assigns each ground-truth object to exactly one query, while all remaining queries are treated as background.

Despite its conceptual elegance, DETR suffers from slow convergence and limited ability to detect small objects. Deformable DETR (D-DETR) (Zhu et al., 2021) alleviates these issues through multi-scale deformable attention, replacing dense global attention with sparse adaptive sampling.

**Sparsemax.** The sparsemax transformation was originally proposed by Martins & Astudillo (2016) in the context of natural language processing (NLP) tasks as an alternative to the softmax function, particularly for multi-label text classification and sequence modeling. Unlike softmax, which always produces dense distributions, sparsemax projects scores onto the probability simplex and can assign exactly zero probability to irrelevant classes, yielding more compact and interpretable outputs. Subsequent works (Maruf et al., 2019; Martins et al., 2021; Ribeiro et al., 2020; Correia et al., 2019) have adopted sparsemax in attention mechanisms, where its sparsity encourages more selective and human-interpretable alignments while preserving performance comparable to softmax. However, to the best of the authors' knowledge, sparsemax has not been employed as the activation function in the final layer of computer vision models, where softmax or sigmoid activations remain standard.

**Hierarchical Taxonomy.** We represent the label space as a directed forest $\mathcal{T} = (N, E)$, where $N$ is the set of nodes , covering the complete hierarchy from coarse-grained (non-leaf) to fine-grained (leaf) classes, and an edge $(a, b) \in E$ indicates that class $a$ is a direct hypernym of class $b$. For a node $c \in N$:

- The *parent* of $c$ is $p(c)$, defined by $(p(c), c) \in E$.
- The *ancestor nodes* of $c$ are

$$\text{Anc}(c) = \{x \in N \mid (x, c) \in E^+\},$$

where $E^+$ is the transitive closure of $E$.

- The set of *leaf nodes* is

$$L = \{c \in N \mid \nexists x \in N : (c, x) \in E\}.$$

- The set of *non-leaf nodes* is $P = N \setminus L$.
- The set of *root nodes* is

$$R = \{c \in N \mid \nexists x \in N : (x, c) \in E\}.$$

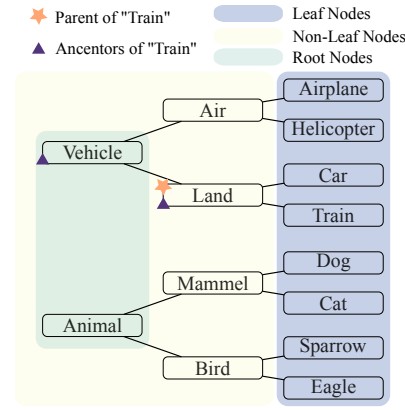

Figure 1: Taxonomy tree serving as an example of the hierarchical structure.

## 3 METHOD

In this section, we present the details of our proposed method (Figure 2). It consists of three main components: an objectness head with sparsemax, a hierarchical-aware activation, and a hierarchy-guided relabeling strategy.

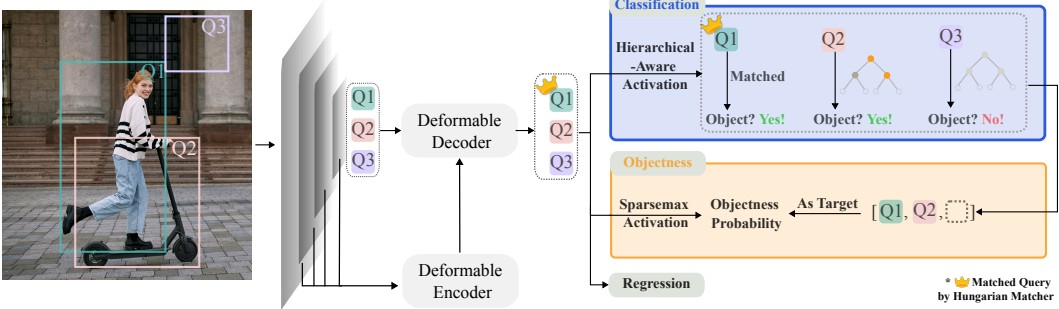

Figure 2: **Overall pipeline of the proposed method.** Building on D-DETR, the classification head first applies a hierarchical-aware activation that couples parent and child classes. Based on these activations, we apply a hierarchy-guided relabeling strategy. In standard D-DETR, only queries that are matched to ground-truth objects by the Hungarian matcher are labeled as positives, with all others treated as background. In contrast, our approach relabels queries with strong non-leaf activations as potential objects (e.g., Q2), while queries with weak activations remain background (e.g., Q3). The target in the objectness head will be updated accordingly, providing auxiliary supervision that complements the primary objectness modeling and helps refine objectness learning. The objectness head uses sparsemax to model competition and sparsity among queries, guided by the updated labels from the classification head. Regression head remains identical as the original D-DETR. For clarity, the Hungarian matcher is omitted, and matched queries are indicated by a crown symbol.

### 3.1 OBJECTNESS WITH SPARSEMAX

**Motivation.** In addition to the classification and localization heads in D-DETR, an objectness head is introduced. This head takes the embeddings from the last decoder layer and produces a scalar score for each query, with sparsemax applied as the activation. The choice of sparsemax is motivated by two considerations: **(1) Competition rather than suppression.** A straightforward approach would be to use a sigmoid activation with hard binary supervision for each query individually. However, this design forces unknown objects to share the negative target with background

queries, and thus being suppressed. To address this, we propose to reformulate the problem from an individual query perspective to a collective one: all queries within an image compete for objectness probability allocation. Sparsemax naturally facilitates this competition by allocating a probability budget across queries so that known objects are encouraged to have higher scores, while unannotated but plausible queries are not explicitly forced to 0. **(2) Sparsity.** Sparsemax produces sparse probability distributions, i.e., many outputs are exactly zero. This aligns well with the characteristics of query-based object detectors, where most queries capture background regions, while only a few correspond to real objects. The sparsity induced by sparsemax therefore provides a principled way to reflect this underlying nature.

**Sparsemax-based Objectness.**  With the motivations of competition and sparsity established, we now describe sparsemax in the objectness head. Specifically, given the logits $\boldsymbol{z} \in \mathbb{R}^Q$ of all $Q$ queries, sparsemax projects them onto the probability simplex, yielding a sparse probability distribution $\boldsymbol{p}$:

$$\mathrm{sparsemax}(\boldsymbol{z}) = \arg\min_{\boldsymbol{p} \in \Delta^{Q-1}} \|\boldsymbol{p} - \boldsymbol{z}\|^2, \tag{1}$$

where $\Delta^{Q-1}$ is the (Q - 1)-dimensional simplex:

$$\Delta^{Q-1} = \left\{ \boldsymbol{p} \in \mathbb{R}^Q \mid \langle \boldsymbol{1}, \boldsymbol{p} \rangle = 1, \ \boldsymbol{p} \geq \boldsymbol{0} \right\}.$$

This projection produces a probability vector $\boldsymbol{p}$ in which many entries are exactly zero, while the active entries remain positive and sum to one. Unlike softmax, which assigns nonzero probability to every query, sparsemax highlights only a subset of queries as relevant. As a result, the scores also become easier to interpret: non-informative background queries are discarded, while only queries with evidence of objects retain positive values.

For training, all positive queries share the probability budget equally, and the sparsemax loss is used:

$$L_{\mathrm{obj}} = L_{\mathrm{sparsemax}}(\boldsymbol{z}; \boldsymbol{q}) = -\boldsymbol{q}^\top \boldsymbol{z} + \frac{1}{2} \sum_{j \in S(\boldsymbol{z})} (z_j^2 - \tau^2(\boldsymbol{z})) + \frac{1}{2}\|\boldsymbol{q}\|^2, \tag{2}$$

where $\boldsymbol{z}$ represents the vector of logits, $\boldsymbol{q}$ is the target distribution vector, $S(\boldsymbol{z})$ denotes the support set, which is the subset of indices where the sparsemax output is non-zero, and $\tau(\boldsymbol{z})$ is a threshold value that determines the boundary of this support set.

### 3.2 HIERARCHICAL-AWARE ACTIVATION

**Motivation.**  Conventional classification heads treat categories as independent and typically return a single label without considering the hierarchy among classes. Simply adding non-leaf nodes into the vector while still treating outputs independently is insufficient. On the one hand, it neglects the natural coupling between parent and child classes, which can lead to **inconsistent predictions** (e.g., predicting a child but not its parent). On the other hand, enforcing strict coupling, where a child can only be active if its parent is active, removes inconsistencies but introduces **error propagation**: mistakes at higher levels cascade downward and prevent recovery at the leaves. For example, if a *Sparrow* is misclassified into the branch *Mammal* instead of *Bird*, all bird subclasses become inaccessible and the correct label can never be reached. These limitations motivate explicitly encoding the hierarchy into the model during training, with greater emphasis placed on coarse-level nodes.

**Hierarchical-Aware Activation.**  To achieve this, we extend the classification head with a hierarchy-aware activation function. Let $y \in (0, 1)^k$ denote per-class sigmoid activations. For each child class $c$ with parent $p(c)$, we define:

$$\tilde{y} = y_c \cdot (y_{p(c)})^{\alpha_c}, \tag{3}$$

where $\alpha_c$ is a learnable strength parameter. Root nodes remain unchanged ($\tilde{y}_r = y_r$). This multiplicative formulation reinforces hierarchical consistency by coupling children with their parents,

while simultaneously propagating parent errors to descendants, thereby increasing the penalty associated with inaccurate coarse-level predictions and highlighting their importance.

The decision to make $\alpha_c$ learnable arises from the observation that the strength of coupling between a child and its parent is not uniform across the taxonomy. For instance, within the parent *Bird*, a *Sparrow* inherits most of its defining features and therefore requires strong coupling, whereas a *Penguin* diverges from the typical bird shape and thus exhibits weaker coupling in a vision-based system. Assigning each child its own learnable parameter $\alpha_c$ allows the model to capture such variability and adaptively infer the appropriate coupling strength in a data-driven manner.

### 3.3 HIERARCHY-GUIDED RELABELING

**Motivation.** Relabeling can provide auxiliary supervisory signals by leveraging the model's own predictions rather than relying solely on annotated ground truth. The idea is to let high-confidence signals from the classification head act as provisional guidance for the objectness head. Although such pseudo-labels are less precise than human-provided annotations, they enrich supervision by reinforcing queries that appear object-like, thereby refining objectness learning.

**Relabeling.** Our model's classification head is supervised using a class hierarchy. For queries matched to ground-truth instances via Hungarian matching, the supervision target is a multi-hot vector where both the ground-truth leaf class and all its ancestors are assigned a positive label. For unmatched queries, supervision is limited to the leaf nodes, which are all assigned a negative target. This prevents the model from classifying them into specific object categories while allowing ambiguity at higher levels of the hierarchy. In essence, it tells the model, "We are certain this isn't a specific known type of object, but it might belong to a general family of objects."

While the sparsemax-based objectness head plays the primary role in highlighting plausible object queries, we complement it with a lightweight relabeling strategy. Specifically, as induced by our supervision signal, the non-leaf predictions of unmatched queries are not explicitly suppressed. As a result, unmatched queries may still exhibit meaningful confidence at non-leaf levels, which we interpret as potential evidence of unknown objects. To make use of this signal, we define an adaptive confidence threshold based on the minimum non-leaf score among all matched queries in an image. Any unmatched query whose non-leaf prediction exceeds this threshold is relabeled as a candidate unknown, and the target in the objectness loss is adjusted accordingly.

## 4 EXPERIMENT

**Datasets.** The experiments were conducted on the OWOD benchmarks, namely the OWOD Split (Joseph et al., 2021), and the OW-DETR Split (Gupta et al., 2022). Images are taken from the PASCAL-VOC2007/2012 (Everingham et al., 2010) and MS-COCO (Lin et al., 2014) datasets, comprising a total of 80 object classes. In each split, the classes are divided into four non-overlapping subsets, each representing a separate task. During training on a given task, only annotations for the classes belonging to that task are provided. The tasks are presented incrementally: when training on Task 2, for instance, the model must learn the new classes introduced in that task while retaining knowledge of the classes from Task 1. During evaluation, the model is tested on the entire test dataset, where images may contain objects from both previously learned classes and classes that have not yet been introduced. The two benchmarks differ in how classes are distributed across tasks: OW-DETR Split groups semantically similar categories together (e.g., all animals belong to Task 1), OWOD Split distributes classes more evenly. More details are available in Appendix D.

**Metrics.** For known objects, we report the mean Average Precision (mAP), computed over all classes the model has encountered up to the current task. To assess the model's handling of unknown objects, we employ three metrics. Unknown Recall (U-R) measures how many objects from future classes are correctly detected as unknown, where in our model "unknown" refers to predictions that stop at a non-leaf node in the hierarchy. Absolute Open-Set Error (AOSE) quantifies the number of objects from future classes misclassified as belonging to a known class, with lower AOSE indicating a clearer separation between known and unknown categories. Hierarchy Accuracy (HAcc) evaluates whether detected unknown objects are assigned to the correct parent node in the semantic hierarchy. Formally, $\text{HAcc} = \frac{1}{N}\sum_{i=1}^{N}\mathbf{1}(\hat{p}(c_i) = p(c_i))$, where $N$ is the number of detected unknown objects, $c_i$ is the ground-truth class of the $i$-th object, $p(c_i)$ is its ground-truth parent, and $\hat{p}(c_i)$ is the predicted parent. All metrics, apart from AOSE, are reported in percentages.

**Baselines for Comparison.** We evaluate TARO against established methods in OWOD. Our primary baselines are DETR-based approaches, namely OW-DETR (Gupta et al., 2022), PROB (Zohar et al., 2023), and ALLOW-DETR (Ma et al., 2024). Additionally, we report results from Faster R-CNN–based methods, including ORE-EBUI (Joseph et al., 2021), UC-OWOD (Wu et al., 2022b), OCPL (Yu et al., 2022), 2B-OCD (Wu et al., 2022a), and RandBox (Wang et al., 2023b), as informative references.

**Implementation.** We adopt the D-DETR (Carion et al., 2020) architecture with a ResNet-50 backbone (He et al., 2016) pretrained using the DINO self-supervised method (Caron et al., 2021). For incremental learning, we adopt the exemplar replay strategy [1] from PROB (Zohar et al., 2023). The model uses 100 object queries, each with a hidden dimension of 256. Both the transformer encoder and decoder consist of six layers. Training is conducted on four NVIDIA H100 GPUs with a batch size of 8 per GPU. All remaining hyperparameters and details of the learning schedule are available in Appendix B.

**Quantitative Result.** Table 1 presents the quantitative results on both the OWOD and OW-DETR Splits. Across tasks, TARO consistently improves U-R, indicating stronger ability to localize unknown objects. On the OWOD Split, some baselines such as RandBox and ALLOW-DETR achieve higher mAP, but this comes at the cost of substantially lower U-R. On the OW-DETR Split, which was introduced more recently and thus has fewer reported baselines, TARO shows steady gains in both U-R and mAP (with the exception of mAP in Task 1).

A key strength of TARO is its ability to assign previously unseen objects to correct parent categories, a capability missing in existing methods. On the OWOD Split, TARO achieves up to 29.9% HAcc, demonstrating meaningful hierarchical reasoning beyond simply separating knowns from unknowns. However, HAcc on the OW-DETR Split is considerably lower. This is mainly due to the OW-DETR Split design, which groups similar classes within the same task. As a result, the model lacks exposure to certain parent categories. For instance, if a task only includes subclasses under *Animal*, the model has no chance to learn the concept of *Food*. Consequently, even when TARO successfully detects unknown objects in OW-DETR Split, their assignments to parent nodes become arbitrary.

Table 1: **Quantitative results on OWOD and OW-DETR Splits.** We report U-R, mAP, and HAcc, which respectively capture the localization of unknown objects, the detection of known objects, and the hierarchical categorization of unknowns. In Task 4, all classes are treated as known, so U-R and HAcc are omitted. Across both splits, TARO achieves consistently higher U-R while maintaining a strong balance between known and unknown detection. Some methods (e.g., RandBox and ALLOW-DETR) report higher mAP, but this comes at the cost of substantially reduced U-R. Importantly, TARO is the only model capable of categorizing unknowns into the hierarchy. N/A indicates that the corresponding functionality is not supported by the method.

| Method | Task 1 | | | Task 2 | | | Task 3 | | | Task 4 |
|---|---|---|---|---|---|---|---|---|---|---|
| | U-R ↑ | mAP ↑ | HAcc ↑ | U-R ↑ | mAP ↑ | HAcc ↑ | U-R ↑ | mAP ↑ | HAcc ↑ | mAP ↑ |
| **OWOD Split** | | | | | | | | | | |
| ORE-EBUI | 4.9 | 56.0 | | 2.9 | 39.4 | | 3.9 | 29.7 | | 25.3 |
| UC-OWOD | 2.4 | 57.0 | | 3.7 | 31.8 | | 8.7 | 24.6 | | 23.2 |
| OCPL | 8.3 | 56.6 | | 7.6 | 39.1 | | 11.9 | 30.7 | | 26.7 |
| 2B-OCD | 10.5 | 56.4 | N/A | 9.0 | 38.1 | N/A | 11.6 | 29.2 | N/A | 25.8 |
| RandBox | 10.6 | **61.8** | | 6.3 | 45.3 | | 7.8 | **39.4** | | **35.4** |
| OW-DETR | 7.5 | 59.2 | | 6.2 | 42.9 | | 5.7 | 30.8 | | 27.8 |
| ALLOW-DETR | 13.6 | 59.3 | | 10.0 | **45.6** | | 14.3 | 38.0 | | 30.6 |
| PROB | 19.4 | 59.5 | | 17.4 | 44.0 | | 19.6 | 36.0 | | 31.5 |
| Ours: TARO | **20.9** | 57.9 | **29.9** | **20.6** | 44.1 | **15.3** | **22.0** | 36.8 | **28.6** | 32.7 |
| **OW-DETR Split** | | | | | | | | | | |
| ORE-EBUI | 1.5 | 61.4 | | 3.9 | 40.6 | | 3.6 | 33.7 | | 31.8 |
| OW-DETR | 5.7 | 71.5 | N/A | 6.2 | 43.8 | N/A | 6.9 | 38.5 | N/A | 33.1 |
| PROB | 17.6 | **73.4** | | 22.3 | 50.4 | | 24.8 | 42.0 | | 39.9 |
| Ours: TARO | **22.6** | 72.7 | **5.0** | **24.8** | **52.0** | **4.0** | **28.3** | **45.8** | **5.0** | **44.4** |

---

[1] Exemplar replay is an incremental learning strategy that reduces forgetting by reusing a subset of past examples during new tasks. More details can be found in Gupta et al. (2022) and Zohar et al. (2023)

Table 2 reports AOSE on the OWOD Split, which measures how often unknown objects are incorrectly classified as known. A lower AOSE score indicates stronger open-world performance, as the model is less likely to confuse novel objects with familiar ones. The comparison is limited to methods that report AOSE. TARO achieves remarkably low AOSE scores across all three tasks (2250, 1449, 1079), substantially outperforming the next-best method.

**Qualitative Result.** Figure 3 presents a qualitative comparison among OW-DETR, PROB, and our proposed TARO. OW-DETR generally struggles to localize unknown objects and occasionally fails to detect known categories as well.

Table 2: **Unknown object confusion on OWOD Split.** AOSE counts unknown objects misclassified as known. TARO cuts this error by over 50% compared to the next-best method.

| | AOSE ↓ | | |
|---|---|---|---|
| | **Task 1** | **Task 2** | **Task 3** |
| ORE-EBUI | 10459 | 10445 | 7990 |
| UC-OWOD | 9294 | 5602 | 3801 |
| OW-DETR | 10240 | 8441 | 6803 |
| OCPL | 5670 | 5690 | 5166 |
| RandBox | 4498 | 1880 | 1452 |
| PROB | 5195 | 6452 | 2641 |
| Ours: TARO | **2250** | **1449** | **1079** |

For instance, in the first example, both the car and the person are missed, while in the second image a refrigerator is incorrectly hallucinated. PROB achieves more reliable localization, e.g., correctly identifying the lamp in the second image, but it can also confuse known objects with unknowns, e.g., misclassifying the people in the third. In contrast, TARO not only improves the detection of known objects, successfully capturing the oven in the second image, and the skis and the snowboard in the third image, but also categorizes unknowns into semantically meaningful groups. For example, the excavator in the first example is recognized as a *Land Vehicle*, and the spatula in the second example is grouped under *Utensils*. These examples highlight the capability of TARO to move beyond generic *Unknown* labels by providing structured and interpretable categorizations of novel objects, thereby enhancing both robustness and semantic richness in open-world detection.

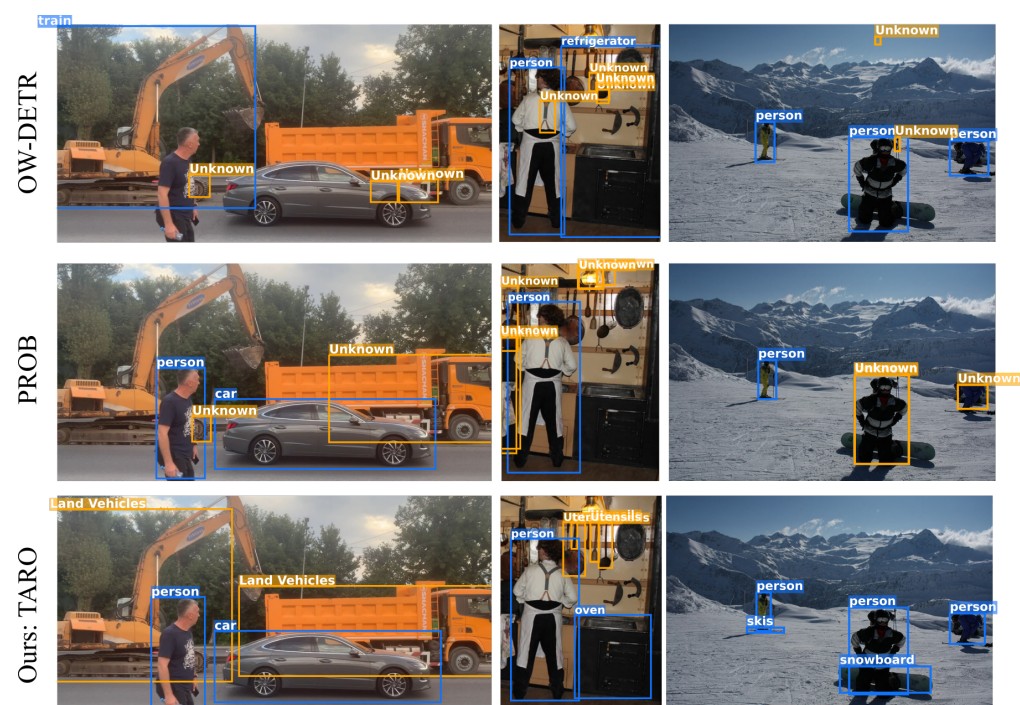

Figure 3: **Qualitative results from TARO (bottom row) compared with OW-DETR (top row) and PROB (middle row).** Predicted known objects are shown in blue, while predicted unknown objects are shown in orange. The first two columns illustrate TARO's capability to detect unknown objects: TARO not only localizes them accurately but also assigns meaningful coarse categories (e.g., the excavator in the first image and the spatula in the second image). The third column highlights TARO's stable performance in detecting known objects. For fair comparison, the same number of top-k predictions is shown for each image. More qualitative results are available in Appendix C.1.

## 5 Ablation Study

To validate the architectural choices and individual contributions of each key component within our proposed TARO model, we conduct a comprehensive ablation study. Experiments are performed on all the tasks from OWOD Split. In this analysis, we systematically deactivate or replace a single component at a time from the full model configuration and evaluate the resulting performance. The following aspects are investigated:

Table 3: **Ablation study of key components in the TARO model on OWOD Split.** Each row measures the performance impact of a specific modification compared to the full model (bottom): (a) *w/ Softmax-Obj:* Replaces the default sparsemax activation with softmax activation in the objectness head. (b) *w/o Relabel:* Disables the relabeling. (c) *w/o L-Strength:* Disables the dynamic learning of hierarchical class relationships.

| Configuration | Task 1 | | | | Task 2 | | | | Task 3 | | | | Task 4 |
|---|---|---|---|---|---|---|---|---|---|---|---|---|---|
| | U-R ↑ | mAP ↑ | HAcc ↑ | AOSE ↓ | U-R ↑ | mAP ↑ | HAcc ↑ | AOSE ↓ | U-R ↑ | mAP ↑ | HAcc ↑ | AOSE ↓ | mAP ↑ |
| w/ Softmax-Obj | 18.1 | 51.7 | 24.8 | 2867 | 17.3 | 35.6 | **18.9** | **1417** | 18.7 | 30.4 | 28.4 | 1210 | 28.2 |
| w/o Relabel | 20.3 | 57.7 | 30.8 | 2263 | 19.7 | 43.6 | 17.6 | 1431 | 20.4 | 36.1 | 26.9 | **882** | 32.4 |
| w/o L-Strength | 19.6 | **58.1** | **31.5** | 2274 | 18.8 | 43.9 | 16.3 | 1497 | 21.3 | **37.0** | **30.7** | 1097 | 32.6 |
| TARO (Full) | **20.9** | 57.9 | 29.9 | **2250** | **20.6** | **44.1** | 15.3 | 1449 | **22.0** | 36.8 | 28.6 | 1079 | **32.7** |

**Sparsemax Objectness.**     To validate our design choice of using sparsemax for the objectness head in TARO, we perform an ablation study where sparsemax is replaced with the more conventional softmax function. Unlike sparsemax, which concentrates probability mass on a subset of high-confidence queries and assigns zero to the rest, softmax distributes non-zero probability across all queries. Following the comparison setup used in the original sparsemax paper (Martins & Astudillo, 2016), softmax is evaluated in combination with the Kullback–Leibler (KL) divergence loss.

As shown in Table 3, replacing sparsemax with softmax causes a substantial drop in both U-R (up to 3.3%) and mAP (up to 8.5%). The difference stems from gradient distribution: with softmax, every query, whether corresponding to a known object, an unknown, or background, receives a non-zero gradient. Since most queries correspond to background, the optimization could be overwhelmed by gradients that suppress background activations, diverting capacity from reinforcing true object signals. In contrast, sparsemax yields sparse distributions where only queries that are part of the support set receive non-zero gradients, enabling the objectness head to focus on positives rather than background suppression. In short, softmax emphasizes turning off negatives, whereas sparsemax emphasizes strengthening positives.

This imbalance also affects the relative weighting between objectness and classification losses. Because the objectness loss under softmax converges less effectively, the classification loss receives less relative weight during training. This slows classification improvements and results in lower mAP compared to the sparsemax-based model.

We also observe higher HAcc and lower AOSE under softmax in Task 2, However, these values are only meaningful when U-R and mAP are comparable. HAcc is computed only over detected unknowns, so when fewer unknowns are recovered the score is evaluated on a reduced subset. For AOSE, the weaker objectness under softmax makes it harder to localize unknowns in the first place, which limits their contribution to this metric. As a result, the lower AOSE is largely a reflection of missed detections rather than improved discrimination.

**Relabel.**     TARO includes a relabeling component that provides auxiliary supervision to the objectness head. In this ablation, we remove this component, so that only explicitly annotated objects in the dataset serve as positive targets in objectness head.

The removal of the relabeling step results in only a minor drop in U-R. This indicates that sparsemax is the primary driver of objectness modeling, while relabeling provides a complementary contribution. By reinforcing high-quality proposals, relabeling refines the model's notion of objectness and enhances robustness, but it functions more as a finetuning technique than a main driver of recall.

Interestingly, removing relabeling also yields a slight but consistent drop in known mAP. This indicates that relabeling helps stabilize learning by mitigating supervision noise. In hand-labeled

datasets like COCO, some known objects are often missing annotations, and relabeling reduces the negative impact of such cases by encouraging the model to trust reliable proposals.

**Learnable Strength.** The classification head of our TARO model incorporates a learnable strength parameter for each node in the class hierarchy. This parameter allows the model to dynamically learn the degree of semantic coupling between a child node and its parent. To validate the contribution of this mechanism, we conduct an ablation study where this learnable parameter is disabled by setting its value to a fixed constant of 0.

Ablating the learnable strength removes the constraint that enforces hierarchical consistency, breaking the coupling between each class and its ancestors at all hierarchical levels. The model then tend to follow the path of least resistance, focusing on the specific and visually distinct features of leaf-node classes while neglecting the more abstract parent categories. This bias reduces its ability to generalize, leading to lower unknown recall. At the same time, H-Acc rises, since the fewer unknowns that are detected tend to resemble known leaf classes and are therefore more easily assigned to the correct branch of the hierarchy.

## 6   FUTURE WORK

While our current relabeling method improves recall for unknown objects, many such objects still go undetected. We believe this reflects a fundamental limitation of image-only OWOD methods: because the notion of "objectness" is learned primarily from known classes, these methods are inherently biased toward them. Consequently, they are more likely to detect unknown objects that share visual similarity with known categories, whereas those with distinct appearances often go unnoticed. A promising future direction is to leverage Vision-Language Models (VLMs). Beyond their broad knowledge, VLMs possess a built-in understanding of semantic hierarchies, making them particularly well suited for guiding relabeling at different taxonomic levels. Incorporating such hierarchical cues could enable more structured pseudo-labels, substantially improving the identification and categorization of unknown objects.

Another future direction could be the utilization of multimodal data to strengthen open world detection. Visual cues alone may be insufficient when unfamiliar objects share only limited visual similarities with known categories, in which case complementary modalities can provide valuable additional information. For example, integrating audio with images could help distinguish rare vehicles such as tractors or excavators by exploiting characteristic engine sounds that resemble those of trucks. Similarly, combining thermal imaging with RGB data can reveal unusual but relevant obstacles, such as small animals crossing the road, which might otherwise be overlooked in standard camera feeds. Such multimodal fusion has the potential to create richer representations, enabling more reliable detection and characterization of unknown objects.

Finally, broader design choices for the taxonomy can be explored. While our current tree structure provides a clean and effective representation, richer forms such as Directed Acyclic Graphs (DAGs) may capture relationships with greater flexibility. Structural properties such as depth and branching factor could also be examined more systematically to understand how they influence hierarchical reasoning. Beyond the choice of structure, it may be valuable to study mechanisms for propagating information within the taxonomy, for example through Graph Neural Networks (GNNs), and to investigate the initialization and learning dynamics of the parent–child coupling parameters. Finally, the possibility of automatically optimizing or even learning the taxonomy may further improve adaptability across different application domains.

## 7   CONCLUSION

This work introduces TARO, a novel and semantically rich framework for OWOD. Moving beyond the conventional *Unknown* label, TARO provides the capability to categorize novel objects into coarse classes, enabling a more informed and nuanced understanding of the open world. The framework is rooted in three strong, interconnected modules: (1) a sparsemax-based objectness head that enforces competition among queries and produces sparse, interpretable objectness scores, (2) a hierarchy-aware activation that enforces consistency across taxonomy levels, and (3) a hierarchy-guided relabeling that leverages non-leaf activations to provide auxiliary supervision for objectness.

Collectively, these three modules enable TARO to deliver strong performance. Experiments on the OWOD and OW-DETR splits show that TARO not only improves unknown recall and significantly reduces confusion between known and unknown objects, but also uniquely provides meaningful coarse-grained categorization of unknowns. By providing richer semantic information about unknown objects, this work extends OWOD beyond a simplistic known–unknown dichotomy.

## 8 REPRODUCIBILITY

To promote transparency and ensure reproducibility, we make the following resources available: (1) **Implementation Details:** Hyperparameters and training settings are documented in the appendix B. (2) **Docker Image:** A fully configured environment will be released upon acceptance, reducing the effort of manually setting up environment while preserving anonymity during review. (3) **Code Repository:** The full implementation is available at `https://anonymous.4open.science/r/TARO`, which also includes training logs and configuration files to facilitate exact reproduction and verification of the reported results.

## 9 THE USE OF LARGE LANGUAGE MODELS (LLMs)

The initial draft of this manuscript was written by the authors. We then utilized LLMs as an assistive tool to improve the grammar, clarity, and overall readability of the text. All suggestions from the LLM were critically reviewed, and the authors made the final decision on all modifications to retain full control over the content. The authors take complete responsibility for the final version of this paper.

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

## A    RELATED WORK

**DETR**    As the first fully end-to-end object detection framework, DETR (Carion et al., 2020) combined a convolutional neural network (CNN) backbone with a transformer architecture. By reformulating detection as a set prediction problem and leveraging bipartite matching loss, DETR eliminated traditional hand-crafted components such as anchor generation and non-maximum suppression (NMS). However, DETR suffered from slow convergence and struggled to localize small objects, mainly due to the computational complexity of global attention and the limited spatial resolution of CNN feature maps. Deformable DETR (D-DETR) (Zhu et al., 2021) addressed these limitations by introducing deformable attention mechanisms, which replaced dense global attention with multi-scale adaptive sparse sampling. While subsequent variants of DETR (Meng et al., 2021; Liu et al., 2022; Li et al., 2022a; Zhang et al., 2023; Dai et al., 2021) further refined query design or training stability, D-DETR remained a foundational advancement for DETR-based framework. Building on it, our work further empowers the model with hierarchical classification and open-world detection capabilities.

**Open Set Recognition (OSR)**    Collecting and labeling a dataset that includes every possible class a model might encounter during deployment is impractical. OSR offers a more feasible solution: instead of requiring annotations for every possible class upfront, OSR trains a model on a limited set of labeled objects. Afterward, during deployment, the model must classify seen objects while identifying unseen ones (Geng et al., 2020).

Saito et al. (2022) proposed BackErase, an augmentation method that pasted annotated objects onto diverse backgrounds, which mitigated the issue of models treating unannotated objects as background. Meanwhile, Jaiswal et al. (2021) tackled foreground-background separation through adversarial training: an auxiliary network was designed between the feature extractor and the bounding box predictor. It predicted object classes from backbone features, but the model would be penalized if the adversarial network succeeded, thereby encouraging the feature extractor to learn class-agnostic features. Furthermore, ALLOW-DETR (Ma et al., 2024) introduces an annealing-based label-transfer framework that avoids manual unknown selection by decoupling known and unknown features and adjusting decision boundaries via a sawtooth annealing schedule. RandBox (Wang et al., 2023b) is a Fast R-CNN–based framework trained on random region proposals instead of relying on biased object distributions. Randomization acts as an instrumental variable that prevents training from being confounded by known categories, while an unbiased matching score encourages exploration of proposals whose predictions do not align with labeled objects. Together, these mechanisms enable the model to better capture unknown instances in the open-world setting.

Other works explicitly formalized foreground detection through a objectness score, which quantified how "object-like" a region is. OW-DETR (Gupta et al., 2022) used backbone feature attention maps to identify regions with high "objectness", correlating high-activation regions with object-like areas, while PROB (Zohar et al., 2023) employed a dedicated regression head to predict objectness scores alongside the traditional classification and localization heads, using the Mahalanobis distance to measure how likely a query embedding represents an object.

Another line of research focuses on using large foundation models to improve open-set and open-world object detection, taking advantage of their strong generalization ability. SGROD (He et al., 2024), for example, uses SAM (Kirillov et al., 2023) to propose all possible objects in an image and then filters out noisy segments so the detector can better identify unknown objects. KTCN (Xi et al., 2024) follows a similar idea: it uses SAM to provide clean pseudo-labels for unknown objects and then introduces a dual-matching strategy to avoid confusion between known and unknown categories. In parallel, OpenDet-D (Luo & Zhou, 2024) combines SAM (Kirillov et al., 2023) and CLIP (Radford et al., 2021) so that SAM supplies objectness information for discovering unknown objects, while CLIP generates visual prototypes that help the detector concentrate on known classes and remain stable when new classes are added.

However, while these methods were able to localize unknowns, they provided limited information beyond labeling regions as *Unknown*. Our work addresses this gap by enabling the categorization of unknown objects.

**Open-Vocabulary Object Detection (OVD)**   OVD addresses a challenge that is related to OSR, since both aim to move beyond a fixed category set. However, the practical goal is different. OVD takes an image together with a text prompt and attempts to localize the object described by that text. Recent advances in OVD are largely driven by Vision–Language Models (VLMs) such as CLIP (Radford et al., 2021), which learn a shared image–text embedding space through large-scale contrastive pre-training.

Current OVD approaches are often categorized into two families: Knowledge Distillation (KD) frameworks and Self-Training or Pseudo-Labeling paradigms.

KD methods focus on explicitly transferring the general recognition ability of pretrained VLMs into the detector. In many KD-based systems, this is achieved by replacing the traditional closed-set classification head with text embeddings from the VLM and training the detector so that region-level visual features align with these text-derived representations. A central challenge in this transfer process is the localization gap. VLMs are optimized for holistic, image-level recognition, which leads to visual features that lack the spatial detail required for accurate region-level localization. Region-based Knowledge Distillation (Gu et al., 2022) addresses this issue by cropping object regions, sending them through the VLM to obtain object-centric embeddings, and training the detector to match its region features to these signals. Other KD-oriented methods extend this idea in different ways. TA-RPN (Choi et al., 2024) introduces text embeddings directly into the Region Proposal Network so that proposal generation is influenced by open-vocabulary semantics rather than solely visual cues. Object-Aware Distillation Pyramid (Wang et al., 2023a) reduces distortion in region-level distillation through adaptive proposal cropping and strengthens knowledge transfer using a simple multi-level distillation pyramid. HD-OVD (Fu et al., 2025) further extends the distillation process by combining instance-level crop alignment, class-level text alignment, and image-level scene alignment.

Complementary to distillation, Self-Training and Pseudo-Labeling paradigms leverage the zero-shot recognition capability of VLMs to generate supervisory signals for unlabeled or partially labeled data, thereby expanding the effective training vocabulary. These approaches typically provide the VLM with a large or predefined vocabulary of candidate class names, obtain predictions for images or region proposals, and treat the resulting class assignments and bounding boxes as pseudo labels. The student detector is then trained on this enlarged annotated set. Notable examples include PB-OVD (Gao et al., 2022), which parses object words from image–caption data and uses VLM attention maps to generate pseudo bounding boxes, and VL-PLM (Zhao et al., 2022), which operates on unlabeled images by first generating region proposals and then using CLIP to assign class labels from a predefined vocabulary, where high-confidence predictions are taken as pseudo labels.

However, OVD still diverges from the setting we target. Its dependence on CLIP exposes the model to a vast and uncontrolled vocabulary during pretraining, making the notion of *Unknown* ambiguous. In addition, OVD assumes that a text prompt is provided at inference, while our problem requires detecting unknown objects without any textual specification or prior semantic information.

**Hierarchical Classification**   Instead of relying on a flat structure, in hierarchical classification, classes are organized into a taxonomy (e.g., "Vehicle" → "Wheeled Vehicle" → "Bicycle"). Early approaches (Cai & Hofmann, 2004; Verma et al., 2012; Beygelzimer et al., 2009) employed non-deep learning methods, such as SVMs or tree-based classifiers, to leverage taxonomic dependencies for improved accuracy. However, recent advances have shifted toward integrating hierarchical structure into deep learning frameworks. For example, Deng et al. (2014) formalized the relationship between different classes using a hierarchy and exclusion graph. Yan et al. (2015) proposed a multi-stage CNN structure, where the shallow layers predicted coarse categories and the deeper layers predicted fine-grained categories.

A critical limitation of the previously mentioned methods was their inability to generalize to novel classes, meaning that they can only perform hierarchical classification for known classes. Addressing this gap, Lee et al. (2018) proposed two strategies: a top-down strategy and a flat strategy. In the top-down strategy, multiple classifiers were developed, and the classification process would stop at a node when the classifier was not confident about its prediction, thereby categorizing the input as a novel object. By contrast, in the flat strategy, the classifier would output a flat probability vector over categories, which also included dummy categories for parent nodes. If the final prediction corresponded to one of these dummy categories, the input was classified as a novel class. To address the lack of training samples for parent nodes, they proposed relabeling and leave-one-out techniques, where some or all images of one or more leaf nodes were relabeled to their parent class. However,

these techniques could confuse the model by training the same samples to represent both the leaf class and the parent class. To address this issue, evidence allocation was introduced (Pyakurel & Yu), where an additional score called "evidence" is proposed to differentiate between known and unknown objects.

While prior hierarchical classification works undeniably advanced the generalization ability of the models, they are restricted to image classification. We propose to extend hierarchical unknown-aware classification to object detection.

## B  IMPLEMENTATION DETAILS

Table 4: General hyperparameters used for all training tasks.

| Parameter | Value |
|---|---|
| Hidden dimension | 256 |
| Number of queries | 100 |
| Batch size | 8 |
| GPU | $4 \times$ NVIDIA H100 (94 GB GPU memory) |
| Base learning rate | 2e-4 |
| Decayed learning rate | 2e-5 |
| Class loss coefficient | 2.0 |
| Objectness loss coefficient | 2.0 |
| Bbox loss coefficient | 5.0 |
| L1 loss coefficient | 2.0 |

Table 5: Training epoch and learning rate schedules.

| Task | Total Epochs | Epoch Range | Learning Rate |
|---|---|---|---|
| **OWOD Split** | | | |
| Task 1 | 50 | 0–39 | $2 \times 10^{-4}$ |
| | | 40–49 | $2 \times 10^{-5}$ |
| Task 2 | 60 | 50–59 | $2 \times 10^{-4}$ |
| Task 2 Finetune | 100 | 60–89 | $2 \times 10^{-4}$ |
| | | 90–99 | $2 \times 10^{-5}$ |
| Task 3 | 115 | 100–114 | $2 \times 10^{-4}$ |
| Task 3 Finetune | 165 | 115–154 | $2 \times 10^{-4}$ |
| | | 155–164 | $2 \times 10^{-5}$ |
| Task 4 | 175 | 165–174 | $2 \times 10^{-4}$ |
| Task 4 Finetune | 240 | 175–229 | $2 \times 10^{-4}$ |
| | | 230–239 | $2 \times 10^{-5}$ |
| **OW-DETR Split** | | | |
| Task 1 | 40 | 0–29 | $2 \times 10^{-4}$ |
| | | 30–39 | $2 \times 10^{-5}$ |
| Task 2 | 50 | 40–49 | $2 \times 10^{-4}$ |
| Task 2 Finetune | 80 | 50–69 | $2 \times 10^{-4}$ |
| | | 70–79 | $2 \times 10^{-5}$ |
| Task 3 | 90 | 80–89 | $2 \times 10^{-4}$ |
| Task 3 Finetune | 120 | 90–109 | $2 \times 10^{-4}$ |
| | | 110–119 | $2 \times 10^{-5}$ |
| Task 4 | 130 | 120–129 | $2 \times 10^{-4}$ |
| Task 4 Finetune | 200 | 130–189 | $2 \times 10^{-4}$ |
| | | 190–199 | $2 \times 10^{-5}$ |

# C  MORE RESULTS

## C.1  MORE QUALITATIVE RESULTS

Figure 4 presents more qualitative results of TARO. As already discussed in the main paper, OW-DETR often struggles to localize unknowns and may even miss some known objects (e.g., the bed in the third image and the leftest person in the fourth). PROB performs better in this regard, successfully detecting unknowns such as the grater in the first image and the ottoman in the third. However, it also shows clear limitations: PROB tends to assign high scores to overlapping unknown boxes (e.g., the piano in the second image and the TV table in the third), and its score ranking is unreliable, with implausible predictions sometimes appearing among the top-k results (e.g., implausible unknown predictions receiving overly high scores in the first three images). In contrast, TARO achieves a better balance. It can detect unknowns while assigning them to appropriate high-level categories, such as labeling the potato and mushrooms as *Fresh Food* in the first image and the coffee table as *Furniture* in the second. At the same time, detections of known objects remain clean and consistent, as shown in the fourth and fifth images. In the last column (TARO's column), the numbers before each label correspond to the prediction indices in Figure 5.

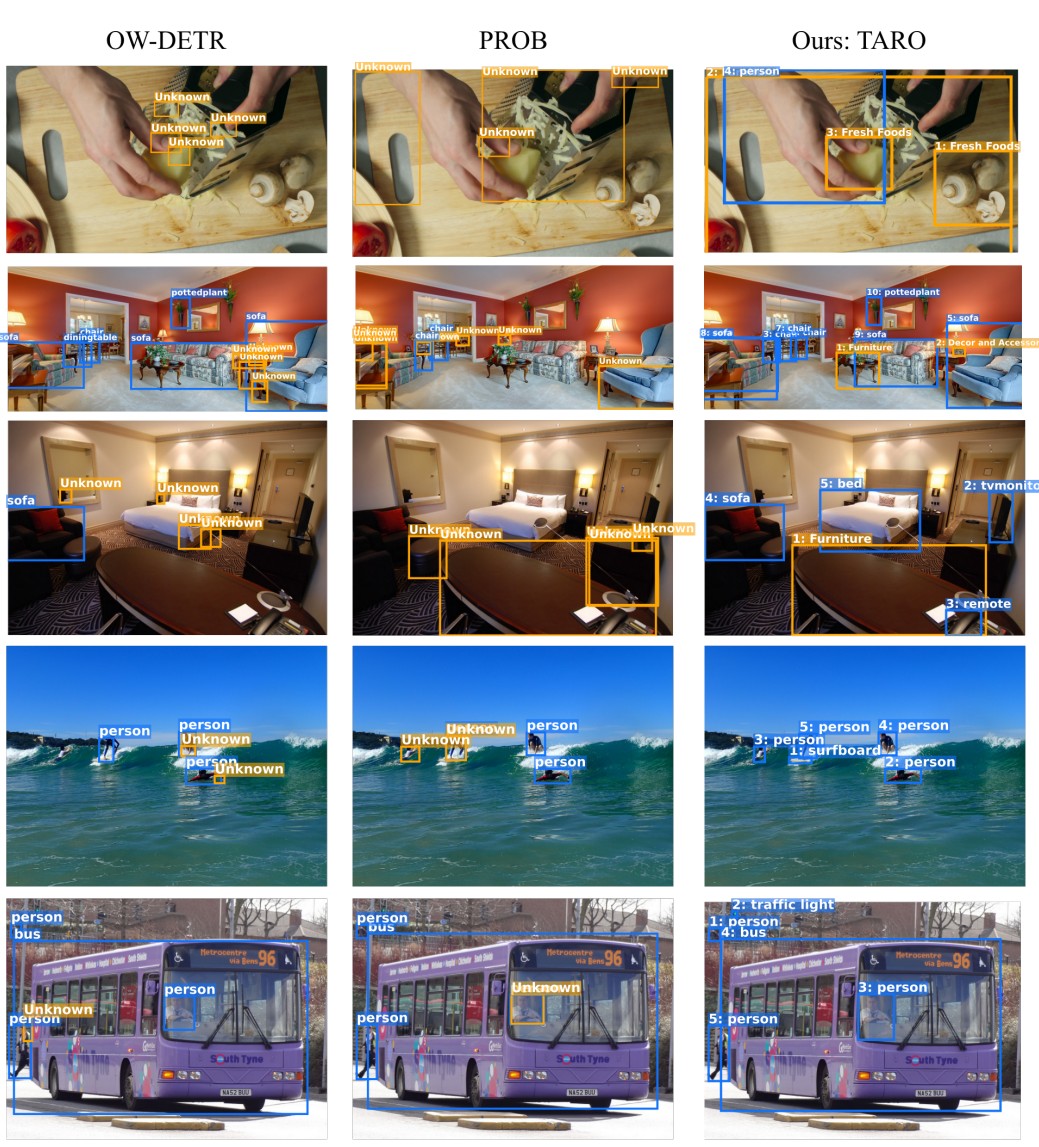

Figure 4: More qualitative result of TARO.

To illustrate TARO's hierarchical reasoning behavior, we provide in Figure 5 the confidence profiles for all predicted objects in Figure 4.

In a hierarchical classifier, coarse-level parent nodes are expected to receive higher confidence than their children. For instance, if the model encounters an unseen *Excavator* after being trained only on *Car* and *Truck*, the desirable behavior is to confidently classify it as a *Vehicle* while remaining uncertain about whether it is specifically a *Car* or a *Truck*.

This behavior, while not explicitly imposed, is encouraged by our hierarchy-aware activation in Eq. 3. The multiplicative coupling biases the model toward satisfying $p(\text{parent}) \geq p(\text{child})$ along each path, while still allowing child nodes to take higher scores when supported by strong visual evidence.

Figure 5 shows bar plots of the predicted confidence across all hierarchy levels. These visualizations demonstrate that TARO tends to follow the expected pattern $p(\text{parent}) \geq p(\text{child})$ for both known and unknown objects. Nevertheless, we note that this relationship could be enforced even more explicitly through additional regularization terms that penalize violations along the hierarchy. Prior work (Li et al., 2022b; Valmadre, 2022; Giunchiglia & Lukasiewicz, 2020) offers promising directions for such extensions.

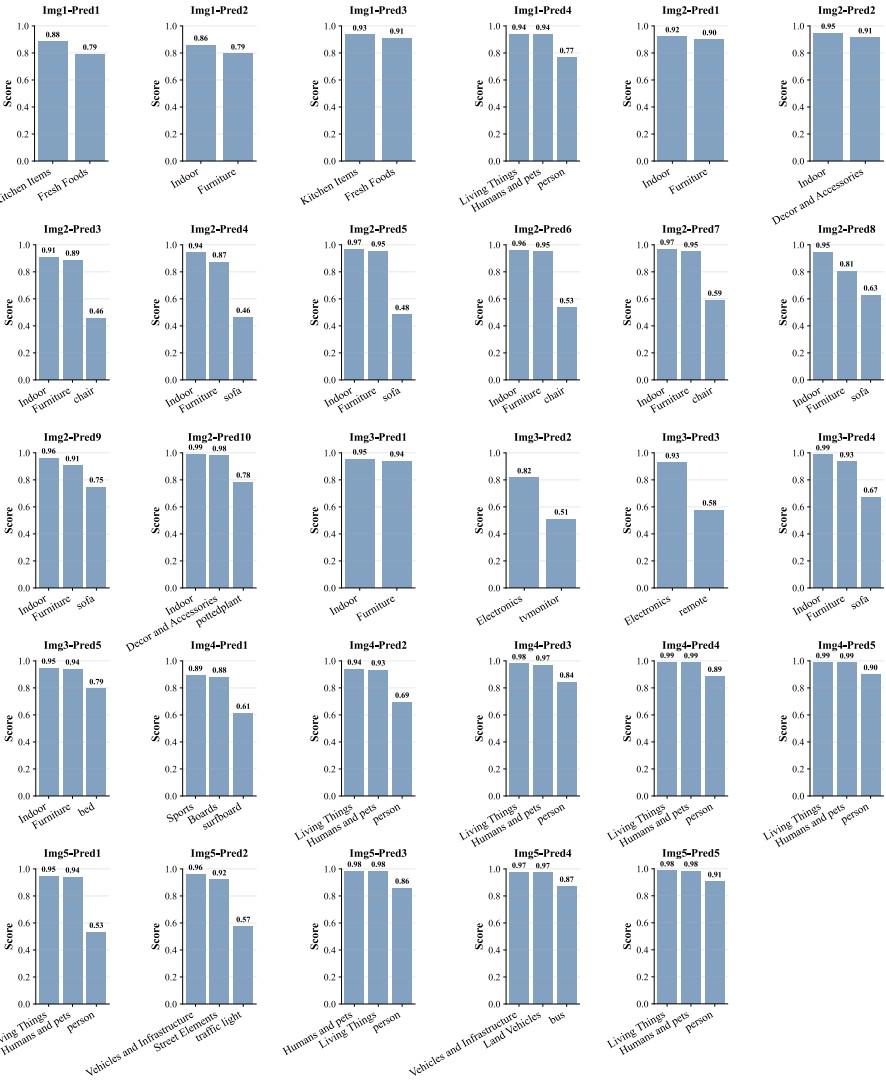

Figure 5: Hierarchical confidence profiles for all predictions. Each subplot corresponds to a predicted object and shows the confidence assigned to the object and its ancestors in the class hierarchy. The label "ImgX–PredY" indicates that the bar chart is derived from the Y-th prediction in the X-th image of Figure 4.

## C.2 MORE QUANTITATIVE RESULTS

Tables 6 and 7 present additional quantitative results of TARO, including Current Known mAP (C-mAP), Previously Known mAP (P-mAP), Both (B-mAP), and Wilderness Impact (WI) (Dhamija et al., 2020). C-mAP is evaluated over classes in the current task, K-mAP over classes introduced in earlier tasks, and B-mAP over all classes introduced until the current task, with B-mAP being the metric reported in the main paper. WI quantifies the performance degradation on known objects caused by interference from unknown objects and is computed as $WI = \frac{FP_u}{TP_k + FP_k}$, where $TP_k$ = true positives for knowns, $FP_K$ = false positives for knowns, $FP_u$ = false positives for unknowns.

We evaluate TARO against established OWOD methods as in the main paper. Our primary baselines are deformable Transformer–based approaches, namely OW-DETR (Gupta et al., 2022), PROB (Zohar et al., 2023), and ALLOW-DETR (Ma et al., 2024). Additionally, we include Faster R-CNN–based methods such as ORE-EBUI (Joseph et al., 2021), UC-OWOD (Wu et al., 2022b), OCPL (Yu et al., 2022), 2B-OCD (Wu et al., 2022a), and RandBox (Wang et al., 2023b) for reference.

As shown in Table 6, TARO generally achieves higher recall compared to all baselines. Some methods obtain higher mAP (e.g., RandBox and ALLOW-DETR), but this comes at the expense of significantly lower unknown recall. Notably, TARO consistently yields much higher current mAP than the competing approaches. Table 7 further demonstrates that TARO maintains a substantially lower AOSE than all other methods. For WI, RandBox consistently ranks first, followed by UC-OWOD in second place and TARO in third.

However, as previously discussed in (Zohar et al., 2023) and (Liang et al., 2023), the WI metric has notable limitations. Models that produce more predictions for known classes naturally increase the denominator $TP_k + FP_k$. This dilutes the impact of false positives from unknown objects $FP_u$, leading to artificially lower WI values. As a result, WI tends to favor models with more known-class predictions rather than those that genuinely handle unknown objects more effectively.

Table 6: More quatitative result on OWOD Split and OW-DETR Split.

| Method | Task 1 | | Task 2 | | | | Task 3 | | | | Task 4 | | |
|---|---|---|---|---|---|---|---|---|---|---|---|---|---|
| | U-R↑ | C-mAP↑ | U-R↑ | P-mAP↑ | C-mAP↑ | B-mAP↑ | U-R↑ | P-mAP↑ | C-mAP↑ | B-mAP↑ | P-mAP↑ | C-mAP↑ | B-mAP↑ |
| **OWOD Split** | | | | | | | | | | | | | |
| ORE* | 4.9 | 56.0 | 2.9 | 52.7 | 26.0 | 39.4 | 3.9 | 38.2 | 12.7 | 29.7 | 29.6 | 12.4 | 25.3 |
| UC-OWOD | 2.4 | 57.0 | 3.7 | 33.1 | 30.5 | 31.8 | 8.7 | 28.8 | 16.3 | 24.6 | 25.6 | 15.9 | 23.2 |
| OCPL | 8.3 | 56.6 | 7.6 | 50.6 | 27.5 | 39.1 | 11.9 | 38.7 | 14.7 | 30.7 | 30.7 | 14.4 | 26.7 |
| 2B-OCD | 10.5 | 56.4 | 9.0 | 47.6 | 27.9 | 38.1 | 11.6 | 37.2 | 13.2 | 29.2 | 30.0 | 13.3 | 25.8 |
| RandBox | 10.6 | **61.8** | 6.3 | - | - | 45.3 | 7.8 | - | - | **39.4** | - | - | **35.4** |
| OW-DETR | 7.5 | 59.2 | 6.2 | 53.6 | 33.5 | 42.9 | 5.7 | 38.0 | 15.8 | 30.8 | 31.4 | 17.1 | 27.8 |
| ALLOW-DETR | 13.6 | 59.3 | 10.0 | - | - | **45.6** | 14.3 | - | - | 38.0 | - | - | 30.6 |
| PROB | 19.4 | 59.5 | 17.4 | **55.7** | 32.2 | 44.0 | 19.6 | **43.0** | 18.2 | 36.0 | 35.7 | 18.9 | 31.5 |
| Ours: TARO | **20.9** | 57.9 | **20.6** | 53.3 | **34.9** | 44.1 | **22.0** | 41.7 | **26.9** | 36.8 | **36.4** | **21.6** | 32.7 |
| **OW-DETR Split** | | | | | | | | | | | | | |
| ORE* | 1.5 | 61.4 | 3.9 | 56.5 | 26.1 | 40.6 | 3.6 | 38.7 | 23.7 | 33.7 | 33.6 | 26.3 | 31.8 |
| OW-DETR | 5.7 | 71.5 | 6.2 | 62.8 | 27.5 | 43.8 | 6.9 | 45.6 | 24.9 | 38.5 | 38.2 | 28.1 | 33.1 |
| PROB | 17.6 | **73.4** | 22.3 | **66.3** | 36.0 | 50.4 | 24.8 | 47.8 | 30.4 | 42.0 | 42.6 | 31.7 | 39.9 |
| Ours: TARO | **22.6** | 72.7 | **24.8** | 60.3 | **44.5** | **52.0** | **28.3** | **48.2** | **41.0** | **45.8** | **46.0** | **39.4** | **44.4** |

Table 7: Unknown object confusion on OWOD Split.

| | Task 1 | | Task 2 | | Task 3 | |
|---|---|---|---|---|---|---|
| | AOSE↓ | WI↓ | AOSE↓ | WI↓ | AOSE↓ | WI↓ |
| ORE-EBUI | 10459 | 0.0621 | 10445 | 0.0282 | 7990 | 0.0211 |
| UC-OWOD | 9294 | 0.0136 | 5602 | 0.0116 | 3801 | 0.0073 |
| OW-DETR | 10240 | 0.0571 | 8441 | 0.0278 | 6803 | 0.0156 |
| OCPL | 5670 | 0.0423 | 5690 | 0.0220 | 5166 | 0.0162 |
| RandBox | 4498 | **0.0240** | 1880 | **0.0078** | 1452 | **0.0054** |
| PROB | 5195 | 0.0569 | 6452 | 0.0344 | 2641 | 0.0151 |
| Ours: TARO | **2250** | 0.0406 | **1449** | 0.0176 | **1079** | 0.0115 |

## D    Hierarchical Taxonomy and Splits

The taxonomy used in training is shown in Figure 6. It is constructed from WordNet (Fellbaum, 1998) with a small amount of human intervention. The raw WordNet hierarchy cannot be used directly because some dataset categories correspond to multiple possible meanings in WordNet. For example, the COCO category *Toaster* may refer to the household appliance or to a person proposing a toast, and the category *Kite* may refer to the toy that flies in the sky or to a fraudulent bill. To address this, we utilize a tool that automatically detects such ambiguities and prompts the user for clarification. For different tasks or use cases, a new taxonomy can therefore be generated from WordNet using this tool with minimal manual input.

More broadly, the notion of an *Object* is inherently ambiguous and highly dependent on the intended application. Even seemingly simple categories can be interpreted in different ways. For instance, *Cup* may be defined by its function (a container), its geometry (a concave shape), or its material (ceramic). As a result, different real-world systems may adopt fundamentally different hierarchical structures. For instance, a household robot may group objects by function, and an autonomous-driving system may group them by dynamic behavior.

The COCO dataset, however, is designed for general-purpose use and therefore contains categories from many unrelated domains, for instance, *Broccoli* and *Airplane*. While this broad coverage is useful for benchmarking, it complicates the construction of a consistent hierarchy. For practical deployments of TARO, the taxonomy would typically need to be redefined to include only task-relevant objects and to reflect the semantics most meaningful for the target application.

The OWOD Split and OW-DETR split is shown in Table 8 with different colors representing different parent nodes.

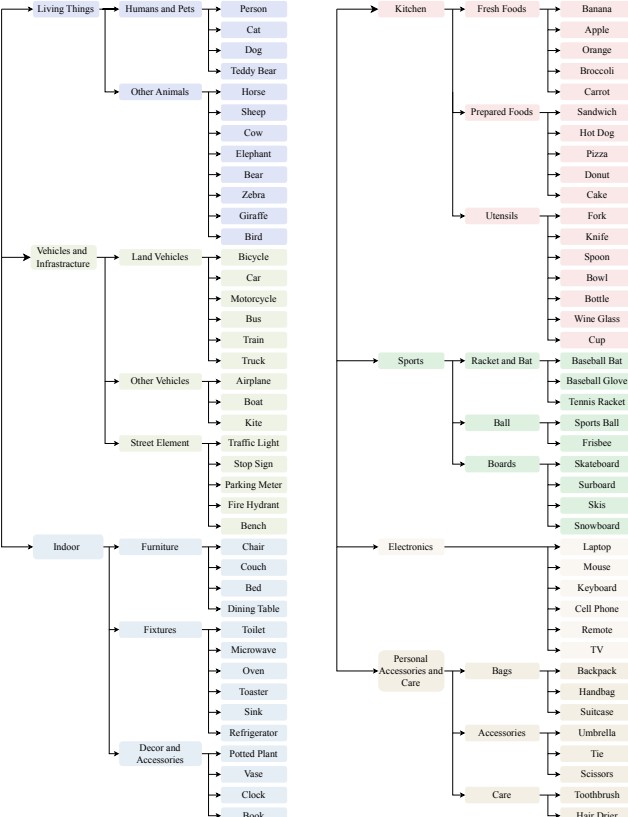

Figure 6: Taxonomy of object categories used in TARO.

Table 8: Classes in OWOD split (Joseph et al., 2021) and OW-DETR split (Gupta et al., 2022) across tasks.

**OWOD Split**

| T1 | T2 | T3 | T4 |
|---|---|---|---|
| Aeroplane | Truck | Frisbee | Bed |
| Bicycle | Traffic light | Skis | Toilet |
| Bird | Fire hydrant | Snowboard | Laptop |
| Boat | Stop sign | Sports ball | Mouse |
| Bottle | Parking meter | Kite | Remote |
| Bus | Bench | Baseball bat | Keyboard |
| Car | Elephant | Baseball glove | Cell phone |
| Cat | Bear | Skateboard | Book |
| Chair | Zebra | Surfboard | Clock |
| Cow | Giraffe | Tennis racket | Vase |
| Diningtable | Backpack | Banana | Scissors |
| Dog | Umbrella | Apple | Teddy bear |
| Horse | Handbag | Sandwich | Hair drier |
| Motorbike | Tie | Orange | Toothbrush |
| Person | Suitcase | Broccoli | Wine glass |
| Pottedplant | Microwave | Carrot | Cup |
| Sheep | Oven | Hot dog | Fork |
| Sofa | Toaster | Pizza | Knife |
| Train | Sink | Donut | Spoon |
| Tvmonitor | Refrigerator | Cake | Bowl |

**OW-DETR Split**

| T1 | T2 | T3 | T4 |
|---|---|---|---|
| Aeroplane | Traffic light | Frisbee | Laptop |
| Bicycle | Fire hydrant | Skis | Mouse |
| Bird | Stop sign | Snowboard | Remote |
| Boat | Parking meter | Sports ball | Keyboard |
| Bus | Bench | Kite | Cell phone |
| Car | Chair | Baseball bat | Book |
| Cat | Diningtable | Baseball glove | Clock |
| Cow | Pottedplant | Skateboard | Vase |
| Dog | Backpack | Surfboard | Scissors |
| Horse | Umbrella | Tennis racket | Teddy bear |
| Motorbike | Handbag | Banana | Hair drier |
| Sheep | Tie | Apple | Toothbrush |
| Train | Suitcase | Sandwich | Wine glass |
| Elephant | Microwave | Orange | Cup |
| Bear | Oven | Broccoli | Fork |
| Zebra | Toaster | Carrot | Knife |
| Giraffe | Sink | Hot dog | Spoon |
| Truck | Refrigerator | Pizza | Bowl |
| Person | Bed | Donut | Tvmonitor |
|  | Toilet | Cake | Bottle |
|  | Sofa |  |  |

Table 9: Number of training images in OWOD and OW-DETR splits across tasks.

| Split | Task 1 | Task 2 | Task 3 | Task 4 |
|---|---|---|---|---|
| **OWOD** | 16551 | 45520 | 39402 | 40260 |
| **OW-DETR** | 89490 | 55870 | 39402 | 38903 |

# E    SCALABILITY ANALYSIS ON LVIS

To further evaluate the generalization and scalability of TARO, we conducted a preliminary study on the LVIS dataset (Gupta et al., 2019). With approximately 1,200 object categories, LVIS provides a substantially larger and more diverse label space than COCO, making it a suitable testbed for examining scalability to deeper and broader taxonomies.

We constructed a hierarchical taxonomy for LVIS using the same procedure described in Supplementary Material D. We then randomly split all categories into two disjoint sets: two-thirds of the classes were made visible to the model during training (the *known set*), and the remaining one-third were withheld entirely (the *unknown set*). During evaluation, these withheld classes were treated as unknown objects.

We analyzed the computational overhead of scaling TARO to the larger LVIS taxonomy. When moving from COCO to LVIS, the training time for the same number of images increased by only 15.5%. In addition, increasing the hierarchy depth by two levels (from 3 to 5) result in an overhead of approximately 5%.

Quantitative results are presented in the Table 10. Both the baseline (PROB (Zohar et al., 2023)) and TARO were trained for 60 epochs. The evaluation metrics, mAP, U-R, and HAcc, follow the definitions in the main paper. We additionally report mAP-COCO, the mAP on the subset of LVIS classes within the known set that overlap with COCO categories (the class mapping between COCO and LVIS follows (Amato et al., 2021)). This is included because LVIS exhibits a pronounced long-tail distribution. For example, 40% of the categories contain fewer than 50 annotated instances across 100170 images, and this trend is clearly visible in Figure 7. Such extreme imbalance results in a low overall mAP. Reporting mAP-COCO provides a more meaningful comparison to Table 1 in the main paper.

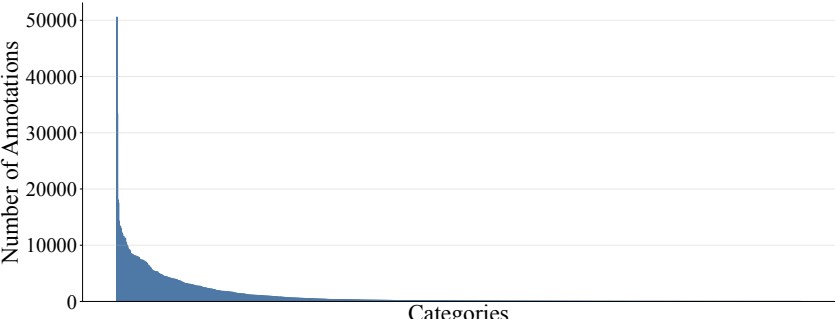

Figure 7: Long-tail distribution of category annotations in the LVIS dataset. Categories are sorted by annotation count.

As shown in Table 10, when scaling from COCO to LVIS, The baseline PROB's ability to detect known objects collapses, reflecting its difficulty in handling the much larger and more imbalanced label space. In contrast, TARO maintains stable performance for both known objects (mAP-COCO) and unknown objects (U-R) when compared to the results reported in Table 1 on COCO, even though no training hyperparameters were changed for LVIS. For HAcc, the results follow an intuitive pattern. When the hierarchy becomes deeper, assigning predictions to the correct parent node becomes more challenging. This reflects a natural trade-off: deeper trees provide more fine-grained semantic information, whereas shallower trees make parent-node assignment easier.

Table 10: Quantitative results on the LVIS scalability experiment.

| Method | mAP | mAP-COCO | U-R | HAcc |
|---|---|---|---|---|
| PROB | 2.9 | 11.5 | 30.1 | – |
| TARO (depth = 3) | 12.0 | 38.8 | 26.8 | 79.5 |
| TARO (depth = 5) | 12.0 | 38.7 | 26.3 | 38.1 |

Figure 8 presents qualitative results on the LVIS dataset. The behavior of the baseline PROB aligns with the quantitative findings: detections for known objects largely collapse. In contrast, TARO demonstrates strong generalization to the LVIS setting. Known objects remain detectable despite the large increase in category count, and the model produces meaningful coarse-level predictions for unseen categories. For example, fried eggs are assigned to the coarse category *food_stuff* in the first image, wooden structures to *material* in the second image. These examples show that TARO scales to a substantially richer taxonomy while maintaining meaningful and interpretable predictions for both known and unknown objects.

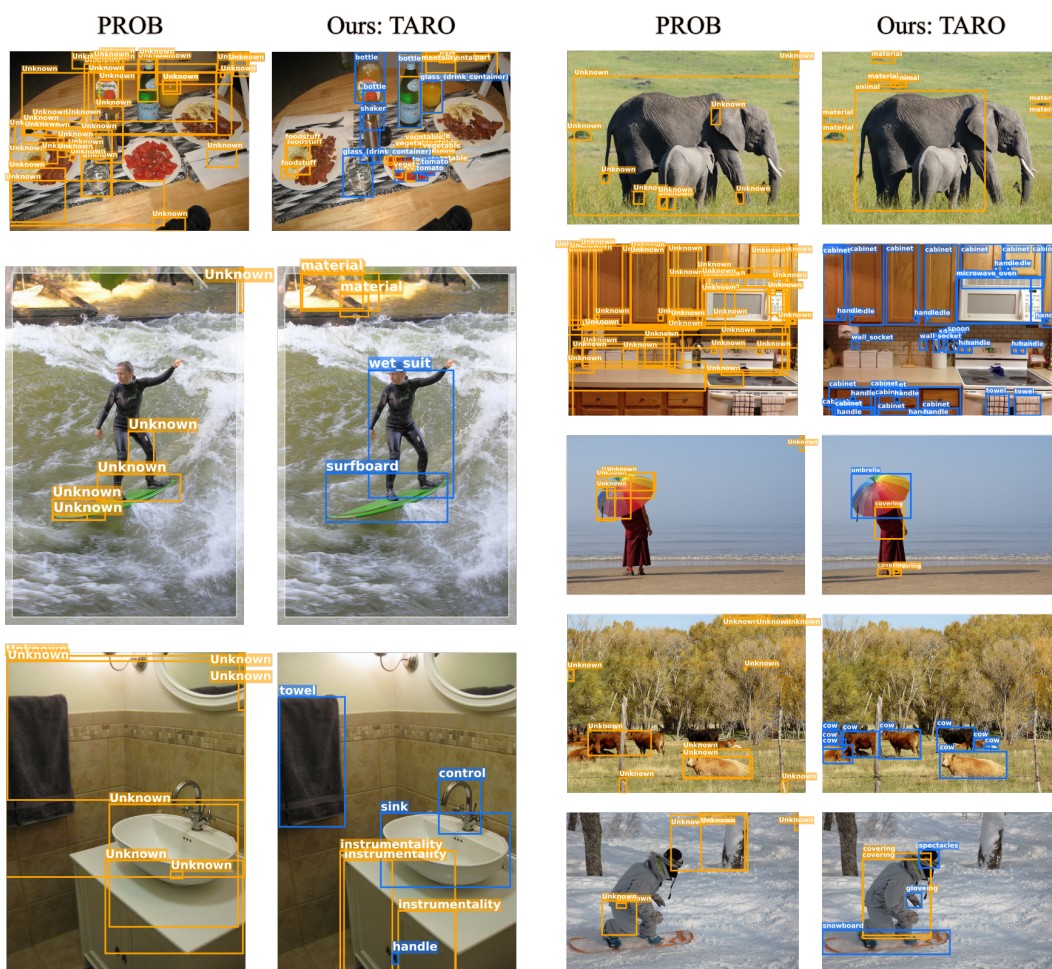

Figure 8: Qualitative result when trained on LVIS dataset. Predicted known objects are shown in blue, while predicted unknown objects are shown in orange.

## F SPARSEMAX GRADIENT STABILITY

When employing a sparse activation such as Sparsemax, a natural concern is whether it could exhibit behavior analogous to the "dead ReLU" problem. This section clarifies why sparsemax does not suffer from this issue.

In the dead ReLU scenario, a neuron develops a strongly negative bias $b_i$, causing its pre-activation $z_i = \mathbf{w}_i^\top \mathbf{x}_i + b_i$ to remain consistently below zero, which results in a permanent zero gradient. Sparsemax is different from ReLU, it is not a fixed switch, it is a competition for every new piece of data.

First, even if a query receives no gradient from the objectness head in a given iteration, the backbone and encoder–decoder layers still receive gradients from other loss components, and the input in the next iteration also differs. As a result, a query having low logits in one batch does not imply it will

have low logits for all future inputs.

Second, the condition for a neuron to enter the sparsemax support set is not governed by a static value such as zero in ReLU, but is determined by its relative standing within the competitive context of the current batch. This threshold is fluid and recalculated for every piece of data, meaning a logit that fails to enter the support in one iteration may successfully do so in the next.

For a more formal and mathematical treatment of Sparsemax support dynamics, readers are referred to the original Sparsemax paper (Martins & Astudillo, 2016).

We also provide here some empirical evidence supporting the stability of the Sparsemax head:

- Throughout the entire training process, no exploding or NaN gradients were observed.

- Quantitative: For a finer-grained analysis, we recorded the gradient magnitudes at the layer where sparsemax was applied. Across all neurons, 0 query exhibited more than 10 consecutive iterations of zero gradient, and the maximum zero-gradient streak among all queries was 3 iterations.

- Qualitative: We additionally provide a gradient heatmap for the first three epochs. No neuron shows a long continuous white band in the heatmap, indicating that none became inactive for extended periods.

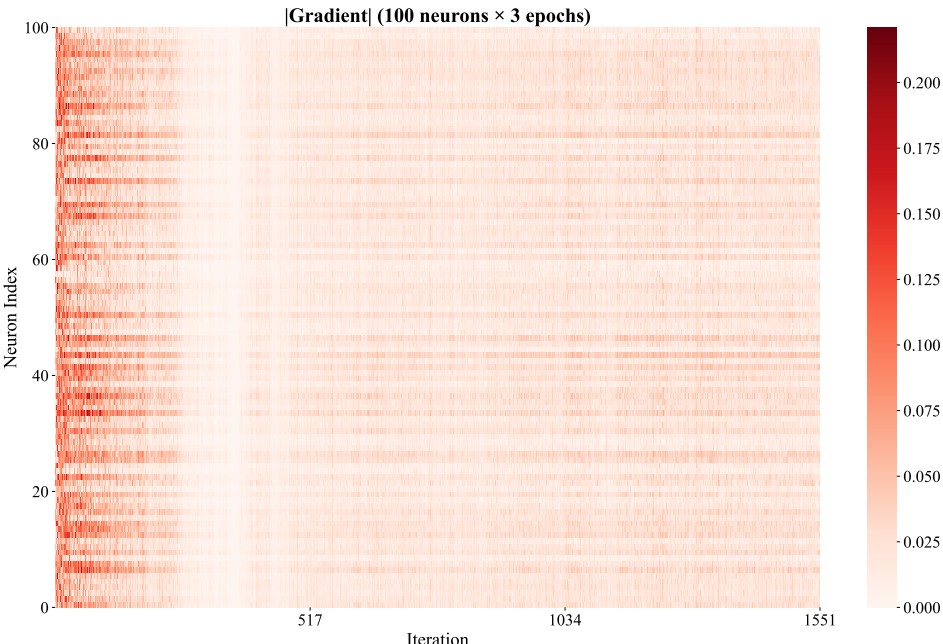

Figure 9: Absolute gradient magnitudes for the neurons in the layer where sparsemax is applied, shown over training iterations during the first three epochs.

