# OpenReview forum: "TARO: Toward Semantically Rich Open-World Object Detection"
_ICLR.cc/2026/Conference — Submitted to ICLR 2026_

### Official Review · Reviewer_Ec8L · 2025-10-27

**Soundness:** 2
**Presentation:** 2
**Contribution:** 1
**Rating:** 2
**Confidence:** 5

**Summary:**

This paper proposes TARO, a novel framework for semantically rich open-world object detection (OWOD). Unlike conventional OWOD methods that label all novel objects as a single “Unknown” class, TARO leverages a semantic hierarchy to assign unknown objects to meaningful coarse-grained parent categories (e.g., “Unknown Vehicle” or “Unknown Animal”). Built upon Deformable DETR, TARO introduces three key components: (1) a sparsemax-based objectness head to model competition and sparsity among object queries; (2) a hierarchy-aware classification activation that enforces parent–child consistency via learnable coupling parameters; and (3) a hierarchy-guided relabeling strategy that uses non-leaf node confidences to provide auxiliary supervision for objectness. Experiments show that TARO achieves high unknown recall (U-R), low absolute open-set error (AOSE), and reports a new metric, hierarchy accuracy (HAcc), reaching up to 29.9% on the OWOD split.

**Strengths:**

The main strength of this work lies in its redefinition of the OWOD task: instead of treating all unknowns uniformly, TARO enables semantically meaningful categorization of novel objects into coarse parent classes, which is highly valuable in safety-critical applications such as autonomous driving. The proposed components, sparsemax-based objectness modeling, adaptive hierarchical coupling, and hierarchy-guided relabeling, are well-motivated and collectively address key challenges in semantic generalization under open-world settings.

**Weaknesses:**

- Limited generalization due to semantic coverage in training: The reported HAcc drops dramatically to ~5% on the OW-DETR split (referred to in the manuscript as a more challenging benchmark), compared to 29.9% on the OWOD split. This is because the OW-DETR split groups semantically similar classes (e.g., all animals) into the same task, preventing the model from ever observing certain parent categories (e.g., “Food”) during training. Consequently, TARO cannot meaningfully categorize unknowns from unseen branches of the hierarchy, revealing a strong dependence on the specific task partitioning and limited true open-world generalization.
- Disproportionate complexity versus performance gain: TARO introduces multiple novel components (sparsemax objectness head, hierarchical activation, relabeling mechanism), yet its known-class mAP consistently lags behind strong baselines such as RandBox and ALLOW-DETR across multiple tasks. While it improves U-R and reduces AOSE, this trade-off—sacrificing known detection accuracy for better unknown recall—may not be acceptable in real-world systems where both capabilities are critical. The marginal gains do not sufficiently justify the added architectural complexity.
- Evaluation bias in HAcc: HAcc is computed only over detected unknown objects, ignoring the large number of missed detections (i.e., low U-R). When U-R is low, HAcc is evaluated on a highly selective subset of “easy” unknowns, potentially inflating the perceived hierarchical reasoning ability. Moreover, the stark discrepancy in HAcc between benchmarks (29.9% vs. 5%) undermines the reliability and comparability of this metric, weakening the paper’s main claim.
- Lack of comparison with recent OWOD and foundation-model-based approaches: The paper does not compare against recent methods that leverage large vision-language models (e.g., CLIP, SAM) or other open-vocabulary detectors[1, 2, 3]. Given the rapid progress in this area, it is essential to position TARO relative to both traditional OWOD methods and modern foundation-model-based alternatives. Without such comparisons, the claimed advantages of TARO remain unsubstantiated.

[1] ktcn: enhancing open-world object detection with knowledge transfer and class-awareness neutralization

[2] Recalling Unknowns Without Losing Precision: An Effective Solution to Large Model-Guided Open World Object Detection

[3] Exploring Orthogonality in Open World Object Detection

**Questions:**

Please see the Weaknesses.

---

> ### Author Response · Authors · 2025-11-20
> **Author Response to Reviewer Ec8L (Part 1/2)**
>
> Thank you for your review and for acknowledging the significance of our task formulation and component design. We respond to your comments below.
>
> >[W1] Limited generalization due to semantic coverage in training: The reported HAcc drops dramatically to ~5% on the OW-DETR split (referred to in the manuscript as a more challenging benchmark), compared to 29.9% on the OWOD split. This is because the OW-DETR split groups semantically similar classes (e.g., all animals) into the same task, preventing the model from ever observing certain parent categories (e.g., “Food”) during training. Consequently, TARO cannot meaningfully categorize unknowns from unseen branches of the hierarchy, revealing a strong dependence on the specific task partitioning and limited true open-world generalization.
>
> >[W3] Evaluation bias in HAcc: HAcc is computed only over detected unknown objects, ignoring the large number of missed detections (i.e., low U-R). When U-R is low, HAcc is evaluated on a highly selective subset of “easy” unknowns, potentially inflating the perceived hierarchical reasoning ability. Moreover, the stark discrepancy in HAcc between benchmarks (29.9% vs. 5%) undermines the reliability and comparability of this metric, weakening the paper’s main claim.
>
> Thank you for those comments. Regarding the low HAcc on the OW-DETR split, as you also noted, one of the primary reasons is the design of the dataset split itself. As shown in Table 8 of the manuscript, both splits tend to group classes sharing the same parent category within the same task, and this is more pronounced in OW-DETR. Under such conditions, it is unrealistic to expect a model to correctly categorize unknown objects belonging to semantic groups it has never encountered. For example, a model trained exclusively on animals cannot meaningfully infer the hierarchical position of an apple, since neither the fine-grained class nor its parent category has ever been observed.
>
> We also want to emphasize that in practical deployments, it is unlikely for a system to suddenly encounter entirely unrelated semantic domains. Typically, an application begins with the semantic groups relevant to its use case, and future unknowns tend to be variations within or near those groups. For example, an autonomous driving system is designed to detect traffic participants and obstacles, and objects from unrelated domains, such as fruit or kitchen utensils, are both unlikely to appear and unnecessary to recognize even if they do.
>
> To further support the point that HAcc depends strongly on how classes are partitioned into tasks, we conducted an additional experiment on LVIS [1] (~1,200 classes). When we *randomly* split the taxonomy into two disjoint sets, trained on the first and evaluated HAcc on the second, the model can reach up to 79.5% HAcc. We hope this helps clarify that TARO exhibits much stronger generalization once semantic coverage is more evenly distributed across tasks.
>
> Regarding the evaluation bias, we agree that U-R and HAcc should be interpreted together. U-R measures how many unknown objects the model can identify, and HAcc evaluates how well those detected unknowns are placed within the hierarchy. The two metrics are therefore complementary. We also acknowledge that, although TARO improves unknown recall compared to prior work, the absolute level of U-R remains relatively low. This is a broader limitation of current OWOD models rather than TARO specifically, and we hope that future progress in this area will further increase unknown recall and lead to more reliable open-world perception.

---

> ### Author Response · Authors · 2025-11-20
> **Author Response to Reviewer Ec8L (Part 2/2)**
>
> >[W2] Disproportionate complexity versus performance gain: TARO introduces multiple novel components (sparsemax objectness head, hierarchical activation, relabeling mechanism), yet its known-class mAP consistently lags behind strong baselines such as RandBox and ALLOW-DETR across multiple tasks. While it improves U-R and reduces AOSE, this trade-off—sacrificing known detection accuracy for better unknown recall—may not be acceptable in real-world systems where both capabilities are critical. The marginal gains do not sufficiently justify the added architectural complexity.
>
> Thank you for raising this concern. We fully agree that both known-class accuracy and unknown-object recall are critical for real-world OWOD systems, and we appreciate the opportunity to clarify this point. Although some baselines achieve slightly higher known-class mAP, TARO remains competitive on known classes, with margins of 3.9, 1.5, and 1.2 on the first three OWOD tasks. At the same time, TARO improves unknown recall by 10.3, 10.6, and 14.2 points compared to these models. We see this as strong evidence that TARO improves the overall open-world capability rather than making a simple trade-off between objectives.
>
> Regarding complexity, TARO does include several additional components, but each is lightweight and targeted toward the challenges specific to OWOD. The backbone and transformer encoder–decoder remain unchanged, so the computational cost is essentially unaffected. For reference, the original D-DETR has 39,744,865 trainable parameters, whereas TARO has 39,748,567 trainable parameters. This is an increase of only 3,702 parameters, which corresponds to approximately 0.009% of the total model size. We hope this reassures the reviewer that the added mechanisms are minimal and purpose-driven.
>
>
>
> >[W4] Lack of comparison with recent OWOD and foundation-model-based approaches: The paper does not compare against recent methods that leverage large vision-language models (e.g., CLIP, SAM) or other open-vocabulary detectors. Given the rapid progress in this area, it is essential to position TARO relative to both traditional OWOD methods and modern foundation-model-based alternatives. Without such comparisons, the claimed advantages of TARO remain unsubstantiated.
>
>
>
> Thanks for raising this point. We agree that both (i) OWOD methods that use vision-language models (VLMs) and (ii) Open Vocabulary Object Detection (OVD) are related to our work. However, they differ significantly in rationale, and a direct comparison would be misleading.
>
> The first major difference lies in the use of pretrained VLMs in both of them. They rely on large-scale models such as CLIP [2] or SAM [3], which are pretrained on hundreds of millions of image–text pairs. Consequently, many of the categories treated as “Unknown” in evaluation are already covered during pretraining.
>
> The next key difference shows up at inference time, where OVD relies on external prompts. In OWOD, the model receives only an image and must detect both known and unknown objects without external guidance. In contrast, OVD requires text prompts at inference: the user must specify which objects to look for. This requirement makes OVD less suitable for some real-world scenarios, such as autonomous driving or disaster-response robotics, where not all novel or surprising objects can be anticipated. But we acknowledge that OVD can be advantageous in other application domains. Examples include content-based image retrieval, where users naturally provide text queries, or human-in-the-loop robotics, where commands like “bring me the green cup” are explicitly given.
>
> We appreciate the reviewer’s suggestion and have added a discussion of OVD as well as OWOD methods that use VLMs to the “Related Work” in supplementary material section A. We kindly refer the reviewer to page 15–16 for the updated text.
>
>
> **References**
>
> [1] Agrim Gupta, Piotr Dollar, and Ross Girshick. Lvis: A dataset for large vocabulary instance segmentation. In Proceedings of the IEEE/CVF conference on computer vision and pattern recognition, pp. 5356–5364, 2019
>
>
> [2] Alec Radford, Jong Wook Kim, Chris Hallacy, Aditya Ramesh, Gabriel Goh, Sandhini Agarwal, Girish Sastry, Amanda Askell, Pamela Mishkin, Jack Clark, et al. Learning transferable visual models from natural language supervision. In International conference on machine learning, pp. 8748–8763. PmLR, 2021.
>
> [3] Alexander Kirillov, Eric Mintun, Nikhila Ravi, Hanzi Mao, Chloe Rolland, Laura Gustafson, Tete Xiao, Spencer Whitehead, Alexander C Berg, Wan-Yen Lo, et al. Segment anything. In Proceedings of the IEEE/CVF international conference on computer vision, pp. 4015–4026, 2023.

---

### Official Review · Reviewer_Je6p · 2025-10-27

**Soundness:** 3
**Presentation:** 3
**Contribution:** 3
**Rating:** 6
**Confidence:** 4

**Summary:**

This paper proposes TARO, a framework for open-world object detection that introduces a hierarchical, semantically aware treatment of unknown categories. Instead of labeling all novel objects as “Unknown,” TARO maps them to parent classes within a taxonomy. The method integrates a Sparsemax-based objectness head, hierarchy-aware activation, and a relabeling strategy, leading to stable and interpretable improvements on OWOD and OW-DETR benchmarks.

**Strengths:**

The idea of hierarchical modeling for open-world detection is intuitive and meaningful, addressing a real limitation of prior “flat” OWOD formulations.

The method design is coherent, with each component targeting a specific issue (semantic inconsistency, suppression of unknowns, weak supervision).

Results are strong and comprehensive, showing consistent gains across key metrics (U-R, AOSE, HAcc).

The work is reproducible and does not rely on external large models, which strengthens its engineering value.

**Weaknesses:**

The hierarchical accuracy gains are somewhat limited; the model seems to underuse the taxonomy signal.

The Sparsemax head lacks gradient or stability analysis—some intuition on optimization behavior would be useful.

Performance varies notably between splits, and the paper could analyze failure cases to better understand the taxonomy imbalance.

Missing comparison with recent vision-language detectors, which could help position TARO among broader open-world approaches.

**Questions:**

Hierarchy-aware activation (Eq. 2):
Could you clarify why a multiplicative interaction is chosen to propagate activation along the taxonomy? Have you explored additive or normalization-based alternatives? A brief sensitivity analysis of the scaling factor α₍c₎ would help justify this design.

Sparsemax head behavior:
Sparsemax can yield zero gradients for inactive logits. Did you observe any optimization instability or dead queries in early training? Some gradient statistics or qualitative examples would be helpful.

Relabeling strategy robustness:
The relabeling threshold (minimum non-leaf score among matched queries) might be sensitive to image content. Have you considered adaptive or percentile-based thresholds? An ablation comparing strategies could clarify its stability.

---

> ### Author Response · Authors · 2025-11-20
> **Author Response to Reviewer Je6P (Part 1/3)**
>
> Thank you for your time and for recognizing the strengths of our hierarchical design, empirical results, and reproducibility. We address your concerns below.
>
>
>
> >[W1] The hierarchical accuracy gains are somewhat limited; the model seems to underuse the taxonomy signal.
>
> We appreciate the reviewer’s observation. We would like to clarify that the low HAcc primarily arises from the design of the OWOD and OW-DETR splits, rather than from limited use of the taxonomy.
>
> As shown in Table 8 of the manuscript, both splits tend to group classes sharing the same parent category within the same task, and this is more pronounced in OW-DETR. Under such conditions, it is unrealistic to expect a model to correctly categorize unknown objects belonging to semantic groups it has never encountered. For example, a model trained exclusively on animals cannot meaningfully infer the hierarchical position of an apple, since neither the fine-grained class nor its parent category has ever been observed. In other words, the taxonomy signal simply cannot be leveraged when it is absent.
>
> We also want to emphasize that in practical deployments, it is unlikely for a system to suddenly encounter entirely unrelated semantic domains. Typically, an application begins with the semantic groups relevant to its use case, and future unknowns tend to be variations within or near those groups. For example, an autonomous driving system is designed to detect traffic participants and obstacles, and objects from unrelated domains, such as fruit or kitchen utensils, are both unlikely to appear and unnecessary to recognize even if they do.
>
> To further verify that TARO does utilize the hierarchical signal when the taxonomy is meaningfully represented across tasks, we conducted an additional experiment on LVIS (~1,200 classes). When we *randomly* split the taxonomy into two disjoint sets, trained on the first and evaluated HAcc on the second, the model can reach up to 79.5% HAcc. We hope this reassures the reviewer that TARO does make use of the taxonomy when the training conditions allow it.
>
>
> >[W2 & Q2] The Sparsemax head lacks gradient or stability analysis—some intuition on optimization behavior would be useful.
>
> **Theoretical perspective:**
>
> We believe your concern relates to the “dead ReLU” problem, where a neuron develops a strongly negative bias $b _i$, causing its pre-activation output $z_i = \mathbf{w} _i^\top \mathbf{x}_i + b _i$ to remain consistently below zero, which results in a permanent zero gradient. Sparsemax is different from ReLU; it is not a fixed switch, it is a competition for every new piece of data.
>
> First, even if a query receives no gradient from the objectness head in a given iteration, the backbone and encoder–decoder layers still receive gradients from other loss components, and the input in the next iteration also differs. As a result, a query having low logits in one batch does not imply it will have low logits for all future inputs.
>
> Second, the condition for a neuron to enter the sparsemax support set is not governed by a static value such as zero in ReLU, but is determined by its relative standing within the competitive context of the current batch. This threshold is fluid and recalculated for every piece of data, meaning a logit that fails to enter the support in one iteration may successfully do so in the next.
>
> **Empirical Validation:**
>
> - Throughout the entire training process, no exploding or NaN gradients were observed.
> - Quantitative: For a finer-grained analysis, we recorded the gradient magnitudes at the layer where sparsemax was applied for the first 3 epochs. Across all queries, 0 of them exhibited more than 10 consecutive iterations of zero gradient, and the maximum zero-gradient streak among all queries was 3 iterations.
> - Qualitative: We additionally provide a gradient heatmap for the first three epochs (please see Figure 9 in Supplementary Material Section F). No neuron shows a long continuous white band in the heatmap, indicating that none became inactive for extended periods.

---

> ### Author Response · Authors · 2025-11-20
> **Author Response to Reviewer Je6P (Part 2/3)**
>
> >[W3] Performance varies notably between splits, and the paper could analyze failure cases to better understand the taxonomy imbalance.
>
> The performance differences mainly stem from the characteristics of the dataset splits rather than the model itself.
>
> For mAP, the gap is largely driven by differences in data volume and class composition. For example, OWOD Task 1 contains 16,551 training images, whereas OW-DETR Task 1 has 89,490. The substantially larger data volume in OW-DETR naturally provides the model with broader visual coverage and leads to stronger mAP (dataset statistics have been added in the supplementary material in Table 9). Moreover, Class composition also plays a role: certain categories (e.g., potted plant, hair dryer) are rare, while others (e.g., chair, sofa) are visually similar and easy to confuse. By contrast, classes such as person (high frequency) or zebra (highly distinctive patterns) are much easier to detect. Since OWOD and OW-DETR assign different class sets to each task, this variation in class difficulty also affects the resulting mAP.
>
> For U-R and HAcc, the main differences arise from the split design. Both splits tend to group classes sharing the same parent category within the same task, and this is more pronounced in OW-DETR. Because of this, a model trained on only a few semantic groups in earlier stages inevitably shapes its notion of objectness around those groups. When the model later encounters unknown objects from entirely new semantic domains, it has no prior exposure, which naturally leads to lower unknown recall. A similar effect appears in hierarchical accuracy. If the model has never seen the parent categories associated with these new semantic domains, it cannot map unknown objects to the appropriate coarse-level ancestors, resulting in lower HAcc. We note that in practical deployments, it is unlikely for a system to suddenly encounter entirely unrelated semantic domains.
>
>
> >[W4] Missing comparison with recent vision-language detectors, which could help position TARO among broader open-world approaches.
>
>
> Thanks for bringing this up! Vision–language detectors cover a broad spectrum of tasks. Among them, the two categories most closely related to TARO are (i) OWOD methods that incorporate vision–language models (VLMs) and (ii) Open-Vocabulary Detection (OVD), which is a specific form of vision–language detection. Our response focuses on these two directions. If the reviewer had additional types of vision–language detectors in mind, we would be happy to clarify.
>
> That said, even for these closely related categories, their underlying assumptions differ substantially from the standard OWOD setting, making a direct comparison misleading.
>
> The first major difference lies in the use of pretrained VLMs in both of them. They rely on large-scale models such as CLIP [1] or SAM [2], which are pretrained on hundreds of millions of image–text pairs. Consequently, many of the categories treated as “Unknown” in evaluation are already covered during pretraining.
>
> The next key difference shows up at inference time, where OVD relies on external prompts. In OWOD, the model receives only an image and must detect both known and unknown objects without external guidance. In contrast, OVD requires text prompts at inference: the user must specify which objects to look for. This requirement makes OVD less suitable for some real-world scenarios, such as autonomous driving or disaster-response robotics, where not all novel or surprising objects can be anticipated. But we acknowledge that OVD can be advantageous in other application domains. Examples include content-based image retrieval, where users naturally provide text queries, or human-in-the-loop robotics, where commands like “bring me the green cup” are explicitly given.
>
> We appreciate the reviewer’s suggestion and have added a discussion of OWOD methods that use VLMs as well as OVD to the “Related Work” in supplementary material section A. We kindly refer the reviewer to page 15–16 for the updated text.

---

> ### Author Response · Authors · 2025-11-20
> **Author Response to Reviewer Je6P (Part 3/3)**
>
> >[Q1] Hierarchy-aware activation (Eq. 2): Could you clarify why a multiplicative interaction is chosen to propagate activation along the taxonomy? Have you explored additive or normalization-based alternatives? A brief sensitivity analysis of the scaling factor α₍c₎ would help justify this design.
>
>
> While normalized additive formulations are a natural consideration for hierarchical modeling, they are unfortunately incompatible with TARO's supervision mechanism for unmatched queries.
>
> In TARO, unmatched queries (i.e., queries corresponding to unknown objects or background regions) are handled as follows: all leaf nodes are assigned a supervision value of 0, because an unmatched query cannot correspond to any known class. In contrast, non-leaf nodes are not supervised, as an unmatched query may still belong to a broad category.
>
> A normalized additive formulation, for example,
>
> $$
> p(\text{car}) = \frac{p(\text{vehicle}) + p(\text{land vehicle}) + p(\text{car})}{3},
> $$
>
> conflicts with this design when applied to unmatched queries. Suppose the model has learned *Car* and *Truck* and now encounters an unseen class such as an excavator. The supervision requires the probabilities of all known leaves to be zero, i.e., $p(\text{car}) = 0$ and $p(\text{truck}) = 0$. Under an additive rule, however, setting the leaf activations to zero also forces their ancestors toward zero, since the averaged value cannot reach zero otherwise. This contradicts the intended behavior in TARO, which keeps non-leaf activations unconstrained so that the model can still express uncertainty at broader semantic levels.
>
> A second limitation of additive schemes is that they impose fixed, uniform weights on all ancestors (for example, each node contributing exactly one-third in the example above). In contrast, our multiplicative formulation, equipped with learnable strength parameters $\alpha _c$, allows each child node to adaptively control both *whether* and *how strongly* parent information should influence it.
>
> Finally, the ablation “w/o L-Strength” (Table 3) evaluates this coupling by removing the learnable influence from all parent–child pairs.
>
>
> >[Q3] Relabeling strategy robustness: The relabeling threshold (minimum non-leaf score among matched queries) might be sensitive to image content. Have you considered adaptive or percentile-based thresholds? An ablation comparing strategies could clarify its stability.
>
> We believe there may be a misunderstanding: the threshold in our method is already *adaptive*. For each image, the relabeling threshold is defined as the minimum non-leaf score among all matched queries, which directly reflects the visual complexity and the number of known objects present. That is, images of cluttered scenes with many potential objects produce a lower threshold and therefore trigger more relabeling. In contrast, cleaner or simpler images yield a higher threshold and consequently fewer relabeled queries, which helps maintain precision. This behavior ensures that the relabeling strategy adapts naturally to the content of each image.
>
> We also tested fixed percentile-based relabeling rules (e.g., relabeling the top 10% or 20% of all queries in one image), but the adaptive relabeling strategy provided better overall performance. For completeness, we report the quantitative comparison below (Task 1–4 on OWOD split).
>
> | **Method**      | **T1 mAP ↑** | **T1 U-R ↑** | **T2 mAP ↑** | **T2 U-R ↑** | **T3 mAP ↑** | **T3 U-R ↑** | **T4 mAP ↑** |
> |-----------------|--------------|--------------|--------------|--------------|--------------|--------------|--------------|
> | **Relabel 10%**     | 57.8         | 20.6         | 43.6         | 19.2         | 36.0         | 21.3         | 32.2         |
> |**Relabel 20%**     | **58.2**     | 20.7         | 42.6         | 19.1         | 35.5         | 21.1         | 32.0         |
> | **Adaptive**    | 57.9         | **20.9**     | **44.1**     | **20.6**     | **36.8**     | **22.0**     | **32.7**     |
>
>
> **References**
>
> [1] Alec Radford, Jong Wook Kim, Chris Hallacy, Aditya Ramesh, Gabriel Goh, Sandhini Agarwal, Girish Sastry, Amanda Askell, Pamela Mishkin, Jack Clark, et al. Learning transferable visual models from natural language supervision. In International conference on machine learning, pp. 8748–8763. PmLR, 2021.
>
> [2] Alexander Kirillov, Eric Mintun, Nikhila Ravi, Hanzi Mao, Chloe Rolland, Laura Gustafson, Tete Xiao, Spencer Whitehead, Alexander C Berg, Wan-Yen Lo, et al. Segment anything. In Proceedings of the IEEE/CVF international conference on computer vision, pp. 4015–4026, 2023.

---

### Official Review · Reviewer_Lu4x · 2025-10-30

**Soundness:** 3
**Presentation:** 2
**Contribution:** 3
**Rating:** 6
**Confidence:** 5

**Summary:**

This paper proposes TARO, a hierarchical open-world object detection framework that categorizes unknown objects into coarse semantic classes using a sparsemax-based objectness head and hierarchy-aware mechanisms. The idea is interesting and the results are promising, but the taxonomy construction and visualization of hierarchical predictions need clearer explanation and stronger experimental support.

**Strengths:**

- Research on object detection in open-world scenarios is of great significance, as it addresses the challenges of recognizing unseen or novel categories in real-world environments.
- Investigating hierarchical OOD (out-of-distribution) detection is particularly valuable, as it enhances the interpretability and semantic understanding of the model’s predictions.

**Weaknesses:**

- The authors seem to have omitted discussion of open-vocabulary object detection in the introduction. This task is somewhat different from open-set detection or closed-set detection, and it is also an important and popular branch.
- In the design principles, the uncertainty (or objectness probability) of unknown objects should ideally increase as the semantic hierarchy becomes coarser, that is, higher-level parent categories should correspond to greater uncertainty. However, this principle is not explicitly reflected in the current design formulation.
- The result presentation is insufficient, the paper only shows qualitative detection outputs. A better way to present the results would be to include bar charts illustrating the predicted categories and objectness confidence at different hierarchical levels, which would make the findings more interpretable and clearly demonstrate the model’s hierarchical reasoning behavior.
- The paper does not specify how the hierarchical taxonomy (T) is constructed, is it based on WordNet, ChatGPT-generated relations, or manually defined structures? The transferability of this taxonomy across different datasets should be discussed. Moreover, since the authors emphasize the *open-world* setting, it is unclear whether the hierarchy needs to be continuously updated as new categories emerge, or if it is designed for specific scenarios and fixed applications. The authors should also discuss how different forms or depths of the taxonomy may affect the model’s performance and stability.

**Questions:**

Please see weaknesses.

---

> ### Author Response · Authors · 2025-11-20
> **Author Response to Reviewer Lu4x (Part 1/2)**
>
> Thank you for your review and for acknowledging that our research addresses a meaningful problem! We address your concerns and questions below.
>
>
> >[W1] The authors seem to have omitted discussion of open-vocabulary object detection in the introduction. This task is somewhat different from open-set detection or closed-set detection, and it is also an important and popular branch.
>
>
> Thanks for pointing this out. We agree that Open-Vocabulary Detection (OVD) are related to Open-World Object Detection (OWOD). However, they differ significantly in rationale, and a direct comparison would be misleading.
>
> The first major difference lies in the use of pretrained vision–language models (VLMs) in OVD. Most OVD approaches rely on large-scale models such as CLIP [1], which are pretrained on hundreds of millions of image–text pairs. Consequently, many of the categories treated as “Unknown” in OVD evaluation are already covered during pretraining. The goal in OVD training is therefore not to learn these categories from scratch, but to transfer localization ability from the training classes to a much broader set of objects.
>
> The second key difference appears at inference. In OWOD, the model receives only an image and must detect both known and unknown objects without external guidance. In contrast, OVD requires text prompts at inference: the user must specify which objects to look for. This requirement makes OVD less suitable for some real-world scenarios, such as autonomous driving or disaster-response robotics, where not all novel or surprising objects can be anticipated. But we acknowledge that OVD can be advantageous in other application domains. Examples include content-based image retrieval, where users naturally provide text queries, or human-in-the-loop robotics, where commands like “bring me the green cup” are explicitly given.
>
> We appreciate the reviewer’s suggestion and have added a discussion of OVD to the “Related Work” in supplementary material section A. We kindly refer the reviewer to page 16 for the updated text.
>
>
> >[W2] In the design principles, the uncertainty (or objectness probability) of unknown objects should ideally increase as the semantic hierarchy becomes coarser, that is, higher-level parent categories should correspond to greater uncertainty. However, this principle is not explicitly reflected in the current design formulation.
>
> >[W3] The result presentation is insufficient, the paper only shows qualitative detection outputs. A better way to present the results would be to include bar charts illustrating the predicted categories and objectness confidence at different hierarchical levels, which would make the findings more interpretable and clearly demonstrate the model’s hierarchical reasoning behavior.
>
> We thank the reviewer for these insightful comments. We would like to first clarify that our framework decouples the objectness and classification heads: the objectness head predicts whether a query contains any object, regardless of whether the object is known or unknown, while the hierarchical module operates only on the class scores.
>
> Concerning the design principle, our intended behavior is that, for an unknown object, coarse-level parent nodes should be more confident (and thus less uncertain) than their fine-grained descendants. For example, if the model is presented with an unseen *Excavator* while having been trained only on *Car* and *Truck*, the ideal behavior is to confidently classify it as a *Vehicle* but remain uncertain about whether it is a *Car* or a *Truck*.
>
> This behavior is encouraged in our formulation through the hierarchy-aware activation in Eq. (3). The multiplicative coupling biases the model toward p(parent) ≥ p(child) along the path, while still allowing children to deviate when the visual evidence is strong.
>
> Regarding the presentation of results, we have added bar plots in the supplementary material C.1 (new Fig. 5) showing the predicted confidence at each hierarchy level. From these plots, one can observe that the model follows the trend p(parent) > p(child).
>
> We also agree that one could enforce the p(parent) ≥ p(child) relation even more explicitly through additional regularization terms that penalize violations along the hierarchy. Prior work (e.g., [2-4]) provides promising directions in this regard, and we now also mention this in supplementary material C.1 as a potential extension of our method.

---

> ### Author Response · Authors · 2025-11-20
> **Author Response to Reviewer Lu4x (Part 2/2)**
>
> >[W4] The paper does not specify how the hierarchical taxonomy (T) is constructed, is it based on WordNet, ChatGPT-generated relations, or manually defined structures? The transferability of this taxonomy across different datasets should be discussed. Moreover, since the authors emphasize the open-world setting, it is unclear whether the hierarchy needs to be continuously updated as new categories emerge, or if it is designed for specific scenarios and fixed applications. The authors should also discuss how different forms or depths of the taxonomy may affect the model’s performance and stability.
>
>
> We appreciate this practical question and we would like to clarify that our taxonomy is generated automatically using WordNet [5], with minimal human intervention when necessary. The raw WordNet hierarchy cannot be used directly because some dataset categories correspond to more than one possible meaning in WordNet. For instance, the COCO category *Toaster* could refer to the household appliance or to a person proposing a toast in WordNet. To address this, we utilize a tool that automatically detects such ambiguities and prompts the user for clarification. For different tasks or use cases, a new taxonomy can therefore be generated from WordNet using this tool with only minimal manual input. We appreciate the reviewer highlighting this point, and we have added clarification in Supplementary Material D.
>
> At the same time, your question also leads us to think about whether there are ways to automate this pipeline even further. We believe that integrating VLMs could help eliminate the remaining human intervention. For datasets with public annotation policies (e.g. [6]), models like CLIP [1] could align class descriptions directly with WordNet definitions. In cases where public documentation is unavailable, a VLM such as ChatGPT could potentially analyze retrieved image patches to infer the intended definition of a class automatically.
>
> Regarding the update of the taxonomy, we note that in OWOD, an oracle (either a human annotator or another algorithm) has to assign labels to previously unknown objects. This corresponds to the transition between training tasks, where one stage ends and a new set of classes becomes available. In TARO, the same oracle also determines whether the existing taxonomy needs revision at that point. If the new classes are semantically close to the existing ones, the current taxonomy can be reused without modification. However, if entirely new semantic groups are introduced, the hierarchy should be updated to incorporate them. In practice, for a specific application, we do not expect the taxonomy to change frequently or dramatically over time. Typically, an application begins with the semantic groups relevant to its use case, and future unknowns tend to be variations within or near those groups. For example, an autonomous driving system is designed to detect traffic participants and obstacles, and objects from unrelated domains, such as fruit or kitchen utensils, are both unlikely to appear and unnecessary to recognize even if they do.
>
> Regarding the influence of different taxonomy structures, we did not include a systematic study in the paper. We agree that this aspect should be examined in more detail and have listed it as future work. Nonetheless, we can provide some empirical evidence that supports the scalability of TARO. When moving from COCO (80 classes) to LVIS [7] (~1,200 classes), where both the depth and width of the hierarchy increase substantially, the training time for processing the same number of images increased by only about 15.5%. When we additionally increased the maximum depth of the LVIS taxonomy by two further levels (from 3 to 5), the training time increased by only around 5%. These results indicate that TARO scales well even with significantly larger and deeper taxonomies. Regarding performance variations across different taxonomy configurations, we observe that mAP and U-R remain stable across different depth levels, while HAcc naturally varies with taxonomy depth, since deeper structures introduce more challenging hierarchical decisions. We also include new qualitative results on LVIS, and we kindly refer the reviewer to Supplementary Material E for additional details.

---

> ### Author Response · Authors · 2025-11-20
> **Author Response to Reviewer Lu4x**
>
> **References**
>
> [1] Alec Radford, Jong Wook Kim, Chris Hallacy, Aditya Ramesh, Gabriel Goh, Sandhini Agarwal, Girish Sastry, Amanda Askell, Pamela Mishkin, Jack Clark, et al. Learning transferable visual models from natural language supervision. In International conference on machine learning, pp. 8748–8763. PmLR, 2021.
>
> [2] Liulei Li, Tianfei Zhou, Wenguan Wang, Jianwu Li, and Yi Yang. Deep hierarchical semantic segmentation. In Proceedings of the IEEE/CVF Conference on Computer Vision and Pattern Recognition, pp. 1246–1257, 2022.
>
> [3] Jack Valmadre. Hierarchical classification at multiple operating points. In Advances in Neural Information Processing Systems, vol. 35, pp. 18034–18045, 2022.
>
> [4] Eleonora Giunchiglia and Thomas Lukasiewicz. Coherent hierarchical multi-label classification networks. In Advances in Neural Information Processing Systems, vol. 33, pp. 9662–9673, 2020.
>
> [5] George A. Miller. WordNet: a lexical database for English. Communications of the ACM, pp. 39–41, 1995.
>
> [6] P. Mortimer, R. Hagmanns, M. Granero, T. Luettel, J. Petereit, and H.-J. Wuensche. GOOSE dataset documentation: Labeling policy. Available online at [https://goose-dataset.de/docs/resources/labeling_policy.pdf](https://goose-dataset.de/docs/resources/labeling_policy.pdf), 2024.
>
> [7] Agrim Gupta, Piotr Dollar, and Ross Girshick. Lvis: A dataset for large vocabulary instance segmentation. In Proceedings of the IEEE/CVF conference on computer vision and pattern recognition, pp. 5356–5364, 2019

---

### Official Review · Reviewer_jC4K · 2025-10-31

**Soundness:** 3
**Presentation:** 2
**Contribution:** 2
**Rating:** 4
**Confidence:** 3

**Summary:**

This paper introduces TARO, a framework for open-world object detection (OWOD) that advances beyond simply flagging unknown instances as “unknown,” instead leveraging a semantic hierarchy to classify unknowns into meaningful coarse categories. TARO integrates a sparsemax-based objectness head, a hierarchy-aware classification module, and a taxonomy-guided relabeling strategy. Results on OWOD and OW-DETR splits show that TARO achieves strong recall of unknown objects, reduces confusion between known and unknown categories, and uniquely categorizes unknowns into higher-level nodes of a semantic hierarchy, supported by an extensive quantitative, qualitative, and ablation analysis.

**Strengths:**

- **Extension of Open-World Object Detection (OWOD):** The core contribution of this work is its move beyond the simple "known vs. unknown" binary. TARO not only identifies novel objects but also groups them into meaningful coarse-grained categories. This capability is highly practical for real-world settings like autonomous driving and aligns more closely with how humans handle novel items.
- **A Well-Integrated Architecture:** The architecture cleverly synthesizes three key components : a sparsemax-based objectness head to manage query competition ; a hierarchy-aware activation (with a learnable strength parameter) to enforce parent-child consistency ; and a hierarchy-guided relabeling strategy to provide auxiliary supervision.
- **Thorough Empirical Evaluation:** The experimental section is robust, comparing TARO against strong baselines based on both DETR and Faster R-CNN . The authors use a comprehensive suite of metrics (like mAP, U-R, and HAcc) to demonstrate performance . The qualitative results in Figures 3 and 4 also visually demonstrate the model's effectiveness.
- **Clear Ablation Studies:** The ablation study in Table 3 clearly dissects the model. It systematically dismantles the architecture to validate the individual contributions of the sparsemax head, the relabeling strategy, and the learnable strength parameter, providing strong support for the final design choices .
- **Reproducibility:** The authors provide complete implementation details, including hyperparameters, training schedules, and a public code repository, ensuring the work can be reproduced .

**Weaknesses:**

1. **Limited Novelty:** The paper's main novelty lies in its synthesis of existing ideas, not in a brand new algorithm. While the integration of sparsemax, hierarchical awareness, and relabeling is well-motivated , these individual components are already known in the field. The contribution is more about a careful and effective combination rather than a new algorithmic discovery.
2. **Hierarchy and Dataset Limitations:** The model is constrained by its fixed, hand-defined taxonomy. The hierarchy used is based on COCO/VOC and is only moderately sized. This is a poor fit for real-world applications like autonomous driving, which need to handle much larger or more fluid taxonomies. The paper acknowledges this  but doesn't offer a practical way to scale the method or adapt it to evolving hierarchies.
3. **Incomplete Justification for Sparsemax:** The reasoning for using sparsemax feels incomplete. While the ideas of competition and sparsity make sense , the ablation study (Table 3) doesn't show a massive performance win. The paper only compares it to a standard softmax, which is arguably a weak baseline. Other obvious alternatives, like focal loss or top-k activations, aren't discussed, making it hard to be sure if sparsemax is truly the best choice.
4. **Limited Qualitative Results and Scalability:** The qualitative results are limited and don't address scalability. The figures (Fig. 3 & 4) show only a few selected examples. This makes it hard to judge how robust the categorization really is, especially for ambiguous objects. It's also unclear how the method would perform computationally or accurately if the taxonomy grew to include hundreds of parent nodes.

**Questions:**

1. **Out-of-Taxonomy Generalization:** Can the approach be extended or modified to address cases where unknown objects fall entirely outside the fixed taxonomy (e.g., a truly new semantic domain)? How would you propose to detect/flag such hard unknowns?
2. **Comparison to Open-Vocabulary Approaches:** Why are recent open-vocabulary/region-word alignment methods omitted from the empirical comparison? Could you add them, and do you expect TARO’s hierarchical approach to outperform such models for coarse-level categorization?
3. **Robustness Analysis:** How sensitive is TARO to (a) the taxonomy structure (e.g., deeper trees or more overlapping parent nodes), (b) thresholding choices in relabeling, and (c) the initial value/range for $\alpha_c$? Could the authors provide additional experiments or analysis?
7. **Scalability:** Given the moderate-sized taxonomy in Figure 5, how does TARO’s computational cost scale with substantially deeper or broader semantic trees? Are there practical bottlenecks in the hierarchy-aware activation or relabeling as taxonomy grows?

---

> ### Author Response · Authors · 2025-11-20
> **Author Response to Reviewer jC4K (Part 1/5)**
>
> We truly appreciate the time and effort you’ve dedicated to reviewing our submission. We are glad that the architecture, evaluation, and reproducibility of our work were well received. We address your concerns below:
>
>
> >[W1] Limited Novelty: The paper's main novelty lies in its synthesis of existing ideas, not in a brand new algorithm. While the integration of sparsemax, hierarchical awareness, and relabeling is well-motivated , these individual components are already known in the field. The contribution is more about a careful and effective combination rather than a new algorithmic discovery.
>
> Thank you for the comment. We would like to clarify the novelty of our contribution at two levels.
>
> (1) Conceptual novelty.
>
> Our primary contribution is extending Open-World Object Detection (OWOD) beyond the single *Unknown* label. To the best of our knowledge, this perspective has not appeared in prior work and and provides a new way to view the OWOD formulation.
>
> (2) Technical novelty.
>
> To realize this idea, we introduce several new components. For objectness modeling, existing approaches [1-3] use an independent per-query score. In contrast, we propose a new viewpoint where objectness is modeled as a competition across all queries. The use of sparsemax for this purpose is also new and brings strong empirical benefits. In addition, although hierarchy awareness and relabeling have been explored in related contexts, our method introduces mechanisms specifically designed for TARO, including the dynamic coupling factor and a parent-node-based relabeling signal.
>
> Overall, our method is not a simple combination of existing techniques. It introduces a new problem formulation for OWOD and the mechanisms required to make this formulation effective in practice.
>
>
> >[W2] Hierarchy and Dataset Limitations: The model is constrained by its fixed, hand-defined taxonomy. The hierarchy used is based on COCO/VOC and is only moderately sized. This is a poor fit for real-world applications like autonomous driving, which need to handle much larger or more fluid taxonomies. The paper acknowledges this but doesn't offer a practical way to scale the method or adapt it to evolving hierarchies.
>
> Thank you for asking this practical question. We would like to clarify that the taxonomy is not fully hand-defined. It is constructed automatically using WordNet [4], with a small amount of person-in-the-loop intervention only when necessary. The raw WordNet hierarchy cannot be used directly because some dataset categories correspond to more than one possible meaning in WordNet. For example, the COCO category *Toaster* may refer to the household appliance or to a person giving a toast at a celebration. In such cases, a human needs to disambiguate the intended meaning. To handle such situations, we use a small tool that automatically prompts the user for clarification whenever an ambiguous category is detected.
>
> For different tasks or use cases, a new taxonomy can therefore be generated from WordNet using this tool with only minimal manual input. We appreciate the reviewer highlighting this point, and we have added clarification in Supplementary Material D in the update manuscript.

---

> ### Author Response · Authors · 2025-11-20
> **Author Response to Reviewer jC4K (Part 2/5)**
>
> >[W3] Incomplete Justification for Sparsemax: The reasoning for using sparsemax feels incomplete. While the ideas of competition and sparsity make sense , the ablation study (Table 3) doesn't show a massive performance win. The paper only compares it to a standard softmax, which is arguably a weak baseline. Other obvious alternatives, like focal loss or top-k activations, aren't discussed, making it hard to be sure if sparsemax is truly the best choice.
>
>
> Thank you for raising this point. We actually did experiment with other alternatives during model development, including sigmoid + focal loss.  Unfortunately at the time of submission, we did not have the resources to run it across all four OWOD tasks, which is why it was not included in the paper.  Regarding top-k activations, our understanding is that they are typically used as a post-processing step, since selecting the top-k elements is non-differentiable and therefore cannot be directly integrated into an end-to-end training pipeline. Since submission, we have completed the full sigmoid+focal-loss experiments. As shown in the table below , sparsemax is generally more suitable for this task than both softmax and sigmoid. Softmax performs noticeably worse, particularly in terms of mAP across all four tasks. Sigmoid with focal loss performs closer to sparsemax, but still lags behind by a small margin.
>
> |     | **T1** | **T1**     | **T1**       | **T1**       | **T2** | **T2**     | **T2**       | **T2**       | **T3** | **T3**     | **T3**       | **T3**       | **T4**     |
> |------------------|--------|------------|--------------|--------------|--------|------------|--------------|--------------|--------|------------|--------------|--------------|-----------|
> |  **Configuration**  | **U-R ↑** | **mAP ↑** | **HAcc ↑** | **AOSE ↓** | **U-R ↑** | **mAP ↑** | **HAcc ↑** | **AOSE ↓** | **U-R ↑** | **mAP ↑** | **HAcc ↑** | **AOSE ↓** | **mAP ↑** |
> | w/ Softmax-Obj   | 18.1   | 51.7       | 24.8         | 2867         | 17.3   | 35.6       | **18.9**     | 1417         | 18.7   | 30.4       | 28.4         | 1210         | 28.2      |
> | w/ Sigmoid-Obj   | 20.1   | 57.8       | 28.7         | 2310         | 19.6   | 43.1       | 16.1         | **1255**     | 19.7   | 35.6       | 28.3         | **974**      | 31.3      |
> | **TARO**  | **20.9** | **57.9** | **29.9**     | **2250**     | **20.6** | **44.1** | 15.3         | 1449         | **22.0** | **36.8** | **28.6**     | 1079         | **32.7** |
>
>
>
>
> >[W4] Limited Qualitative Results and Scalability: The qualitative results are limited and don't address scalability. The figures (Fig. 3 \& 4) show only a few selected examples. This makes it hard to judge how robust the categorization really is, especially for ambiguous objects. It's also unclear how the method would perform computationally or accurately if the taxonomy grew to include hundreds of parent nodes.
>
>
>
> Thank you for raising this concern. To address scalability, we did an analysis where TARO is extended from COCO (80 classes) to LVIS (~1200 classes) [5]. This substantially enlarges the taxonomy, yet both performance on known objects and unknown objects remain stable (for detail please see supplementary material section E in the updated manuscript), indicating that TARO scales well even under a much broader and deeper hierarchy. Computationally, moving from COCO to LVIS increases training time by only 15.5% for the same number of images. We also provide additional qualitative results on LVIS (Figure 8 in the updated manuscript), which contains many more fine-grained and ambiguous categories. We hope this reassures the reviewer that TARO maintains both qualitative reliability and computational scalability as the taxonomy grows.

---

> ### Author Response · Authors · 2025-11-20
> **Author Response to Reviewer jC4K (Part 3/5)**
>
> >[Q1] Out-of-Taxonomy Generalization: Can the approach be extended or modified to address cases where unknown objects fall entirely outside the fixed taxonomy (e.g., a truly new semantic domain)? How would you propose to detect/flag such hard unknowns?
>
>
> Thank you for this insightful question! Detecting objects that fall entirely outside the predefined taxonomy would be indeed challenging for any image-only OWOD models. When the model has only been trained on a limited set of classes, its learned notion of “objectness” inevitably becomes biased toward the known classes. This limitation is also reflected in the low HAcc scores on the OWDETR split, where the model struggles to identify objects from unseen semantic groups.
>
> Extending TARO to truly out-of-taxonomy unknowns would likely require additional sources of semantic knowledge. A promising direction, which we are also actively exploring, is to combine vision-language models (VLMs) with traditional vision-based detectors. VLMs inherently possess a broad understanding of the world and can capture relationships between categories, which is particularly valuable especially for our setting.
>
> However, directly relying on VLMs remains difficult because (1) their localization ability is still unreliable and (2) their inference cost is often too high for real-time detection. For these reasons, we believe that hybrid approaches, such as (1) injecting VLM features into the detector or (2) using VLM predictions as pseudo-labels during training for the detectors, are promising directions for detecting hard unknowns that fall completely outside the taxonomy.
>
>
>
> >[Q2] Comparison to Open-Vocabulary Approaches: Why are recent open-vocabulary/region-word alignment methods omitted from the empirical comparison? Could you add them, and do you expect TARO’s hierarchical approach to outperform such models for coarse-level categorization?
>
>
> We agree that Open-Vocabulary Detection (OVD) are related to OWOD. However, they differ significantly in rationale, and a direct comparison would be misleading.
>
> The first major difference lies in the use of pretrained VLMs in OVD. Most OVD approaches rely on large-scale models such as CLIP [6], which are pretrained on hundreds of millions of image–text pairs. Consequently, many of the categories treated as “Unknown” in OVD evaluation are already covered during pretraining. The goal in OVD training is therefore not to learn these categories from scratch, but to transfer localization ability from the training classes to a much broader set of objects.
>
> The second key difference appears at inference. In OWOD, the model receives only an image and must detect both known and unknown objects without external guidance. In contrast, OVD requires text prompts at inference: the user must specify which objects to look for. This requirement makes OVD less suitable for some real-world scenarios, such as autonomous driving or disaster-response robotics, where not all novel or surprising objects can be anticipated. But we acknowledge that OVD can be advantageous in other application domains. Examples include content-based image retrieval, where users naturally provide text queries, or human-in-the-loop robotics, where commands like “bring me the green cup” are explicitly given.
>
> We appreciate the reviewer’s suggestion and have added a discussion of OVD to the “Related Work” in supplementary material section A. We kindly refer the reviewer to page 16 for the updated text.
>
> Regarding coarse-level categorization, this is generally not the focus of OVD methods. To our understanding, OVD plays a role closer to “search”, where the model is guided by a text query and must locate matching objects in the image. As a result, recent OVD methods [7,8] focus more on aligning coarse text prompts with fine-grained visual categories, rather than the other way around. For example, if the user provides the prompt “animal”, the detector should identify dogs, cats, or horses in the image.

---

> ### Author Response · Authors · 2025-11-20
> **Author Response to Reviewer jC4K (Part 4/5)**
>
> >[Q3] Robustness Analysis: How sensitive is TARO to (a) the taxonomy structure (e.g., deeper trees or more overlapping parent nodes), (b) thresholding choices in relabeling, and (c) the initial value/range for? Could the authors provide additional experiments or analysis?
>
> **(a) Taxonomy Structure**
>
> Regarding overlapping parent nodes, TARO is built on a forest of hierarchical trees where each non-root node has exactly one parent, so overlapping parent nodes do not appear in our representation.
>
> Regarding tree depth, COCO contains only 80 classes, which limits the possible variation in hierarchical depth. To study this factor more meaningfully, we conducted experiment on the LVIS dataset (~1,200 classes), which supports deeper and more fine-grained hierarchies. As can be seen from the table below, across different depth configurations, performance on known objects (mAP and mAP-COCO) and unknown objects (U-R) remains stable (mAP-COCO is the mAP on the subset of LVIS classes overlapping with COCO categories, included to account for LVIS’s long-tail distribution. Further details are provided in Supplementary Material E.). The primary change is observed in HAcc. As the hierarchy becomes deeper, assigning predictions to the correct parent node naturally becomes more challenging. This leads to an intuitive trade-off: deeper trees offer more fine-grained information, while shallower trees make parent-node assignment easier.
>
>
> | **Max Depth** | **mAP ↑** | **mAP-COCO ↑** | **U-R ↑** | **HAcc ↑** |
> |---------------|-----------|----------------|-----------|------------|
> | **3**         | 12.0      | 38.8           | 26.8      | 79.5       |
> | **5**         | 12.0      | 38.7           | 26.3      | 38.1       |
>
>
>
> **(b) Relabel Threshold**
>
> TARO does not rely on any manually chosen or fixed threshold. Instead, the threshold is dynamic and image-adaptive: for each image, the threshold is set to the lowest parent-node score among all matched queries. This design ensures that the threshold naturally adjusts to the difficulty and content of each image. For example, in cluttered or visually complex images, parent-node activations tend to be lower because the scene contains more ambiguous signals. In such cases, the threshold also becomes lower and more objects get relabeled. In contrast, in cleaner or simpler scenes, parent-node activations tend to be higher because the model is more certain about the known objects, and the threshold becomes more selective, helping maintain precision.
>
> **(c) Initial Value**
>
> If we understand the reviewer correctly, this refers to the initialization of the coupling strength between parent and child nodes. We initialize this parameter to 1, which provides a simple and interpretable starting point: the child initially inherits its parent activation with unit strength. Since the parameter is learned during training, the model adjusts it based on data, and we observe that it converges to 0.86 ± 0.11. We interpret this behavior as an indication that the chosen initialization lies in a reasonable range. We acknowledge that we did not conduct a detailed study of different initialization ranges in this submission, and we thank the reviewer for raising this point. We have added this aspect to our “Future Work” discussion.
>
>
> >[Q4] Scalability: Given the moderate-sized taxonomy in Figure 5, how does TARO’s computational cost scale with substantially deeper or broader semantic trees? Are there practical bottlenecks in the hierarchy-aware activation or relabeling as taxonomy grows?
>
>
> **Taxonomy Width (Number of Classes):**
>
> A broader taxonomy primarily affects the final MLP in the classification head, whose parameters scale linearly with the number of classes. This introduces minimal overhead, as the computational cost is dominated by the backbone, encoder and decoder, and these components do not depend on the number of classes.
>
> **Taxonomy Depth (Hierarchy Levels):**
>
> A deeper taxonomy impacts the hierarchy-aware activation module, which operates along the ancestor path of each class. The cost therefore scales with the path length rather than the total taxonomy size.
>
> **Empirical Behavior:**
>
> To provide quantitative evidence, when moving from COCO (80 classes) to LVIS (~1200 classes), where both the width and depth increase, the training time for processing the same number of images increased by only about 15.5%. When we further increased the maximum depth of the LVIS taxonomy by two additional levels (from 3 to 5), the training time increased by only ~5%. We hope this helps clarify that TARO offers favorable computational scaling even as the taxonomy becomes much larger.

---

> ### Author Response · Authors · 2025-11-20
> **Author Response to Reviewer jC4K (Part 5/5)**
>
> **References:**
>
>  [1] Akshita Gupta, Sanath Narayan, KJ Joseph, Salman Khan, Fahad Shahbaz Khan, and Mubarak Shah. Ow-detr: Open-world detection transformer. In Proceedings of the IEEE/CVF conference on computer vision and pattern recognition, pp. 9235–9244, 2022.
>
> [2] Orr Zohar, Kuan-Chieh Wang, and Serena Yeung. Prob: Probabilistic objectness for open world object detection. In Proceedings of the IEEE/CVF Conference on Computer Vision and Pattern Recognition, pp. 11444–11453, 2023.
>
> [3] KJ Joseph, Salman Khan, Fahad Shahbaz Khan, and Vineeth N Balasubramanian. Towards open world object detection. In Proceedings of the IEEE/CVF conference on computer vision and pattern recognition, pp. 5830–5840, 2021
>
> [4] George A. Miller. WordNet: a lexical database for English. Communications of the ACM, pp. 39–41, 1995.
>
> [5] Agrim Gupta, Piotr Dollar, and Ross Girshick. Lvis: A dataset for large vocabulary instance segmentation. In Proceedings of the IEEE/CVF conference on computer vision and pattern recognition, pp. 5356–5364, 2019.
>
> [6] Alec Radford, Jong Wook Kim, Chris Hallacy, Aditya Ramesh, Gabriel Goh, Sandhini Agarwal, Girish Sastry, Amanda Askell, Pamela Mishkin, Jack Clark, et al. Learning transferable visual models from natural language supervision. In International conference on machine learning, pp. 8748–8763. PmLR, 2021
>
> [7] Mingxuan Liu, Tyler L Hayes, Elisa Ricci, Gabriela Csurka, and Riccardo Volpi. Shine: Semantic hierarchy nexus for open-vocabulary object detection. In Proceedings of the IEEE/CVF Conference on Computer Vision and Pattern Recognition, pp. 16634–16644, 2024.
>
> [8] Lewei Yao, Jianhua Han, Youpeng Wen, Xiaodan Liang, Dan Xu, Wei Zhang, Zhenguo Li, ChunjingXu, and Hang Xu. Detclip: Dictionary-enriched visual-concept paralleled pre-training for open- world detection. Advances in Neural Information Processing Systems, 35:9125–9138, 2022.

---

### Meta-Review · Area_Chair_dhqB · 2025-12-28

**Summary:**

The paper proposes a framework for open-world object detection that categorizes unknown objects into coarse parent classes using hierarchy-aware designs. Specifically, the authors introduce sparsemax-based objectness, hierarchy-aware activation, and hierarchy-guided relabeling modules to enable classification of unknown instances into coarse parent categories.

Reviewers raised several concerns, including limited novelty, insufficient justification for the use of sparsemax, limited generalization across datasets, inadequate qualitative results, unclear framework design, modest performance gains, missing comparisons with related methods, and insufficient analysis.

While the authors addressed some of these concerns in the rebuttal, several key issues that are critical to strengthening the paper’s contribution remain unresolved. In particular, the justification and analysis for adopting sparsemax are still insufficient, and the generalization ability of the proposed method is not convincingly demonstrated. As a result, the AC believes that the current version of the paper is not yet ready for acceptance and therefore recommends **rejection**.

Nevertheless, the AC acknowledges the potential value of the proposed framework in extending open-world object detection to categorize unknown objects into coarse parent classes. The AC encourages the authors to provide deeper theoretical or empirical analysis to justify the use of sparsemax for this framwork and to more thoroughly address the reviewers’ concerns regarding generalization, especially to unseen parent categories, in future revisions.

**Reviewer Concerns:**

Reviewers raised concerns regarding limited novelty, insufficient dataset generalization, incomplete justification for using sparsemax, inadequate qualitative results, unclear framework design, modest performance gains, lack of in-depth analysis, missing comparisons with related methods, and limited generalization.

Although the authors addressed some reviewer concerns in the rebuttal, several issues critical to strengthening the paper’s contribution remain unresolved. In particular, the justification and analysis for adopting sparsemax are still insufficient, and the generalization ability of the proposed method is not convincingly demonstrated.

**Reviewer Scores:**

Due to unresolved critical concerns, the reviewers are expected to retain their scores: 4, 6, 6, and 2.

---

### Decision · Program_Chairs · 2026-01-26

Reject